



Earth System
Science
Data

# Imputation of missing land carbon sequestration data in the AR6 Scenarios Database

**Ruben Prütz**[1,2,3,4]**, Sabine Fuss**[1,2,4]**, and Joeri Rogelj**[3,5,6]

[1]Geography Department, Humboldt-Universität zu Berlin, Berlin, Germany
[2]Mercator Research Institute on Global Commons and Climate Change (MCC), Berlin, Germany
[3]Grantham Institute for Climate Change and the Environment, Imperial College London,
London, United Kingdom
[4]Potsdam Institute for Climate Impact Research (PIK), Member of the Leibniz Association,
Potsdam, Germany
[5]Centre for Environmental Policy, Imperial College London, London, United Kingdom
[6]Energy, Climate and Environment Program, International Institute for Applied Systems Analysis (IIASA),
Laxenburg, Austria

**Correspondence:** Ruben Prütz (pruetz@mcc-berlin.net)

**Abstract.** The AR6 Scenarios Database is a vital repository of climate change mitigation pathways used in the latest Intergovernmental Panel on Climate Change (IPCC) assessment cycle. In its current version, many scenarios in the database lack information about the level of anthropogenic carbon dioxide ($CO_2$) removal via land sinks, as net-negative $CO_2$ emissions and gross removals on land are not always separated and are not consistently reported across models. This makes scenario analyses focusing on $CO_2$ removal challenging. We test and compare the performance of different regression models to impute missing data on land carbon sequestration for the global level and for several sub-global macro-regions from available data on net $CO_2$ emissions from agriculture, forestry, and other land uses. We find that a $k$-nearest neighbors regression performs best among the tested regression models and use it to impute and provide two publicly available imputation datasets (https://doi.org/10.5281/zenodo.13373539, Prütz et al., 2024) on $CO_2$ removal via land sinks for incomplete global scenarios ($n = 404$) and incomplete regional R10 scenario variants ($n = 2358$) of the AR6 Scenarios Database. We discuss the limitations of our approach, the use of our datasets for secondary assessments of AR6 scenario ensembles, and how this approach compares to other recent AR6 data reanalyses.

## 1 Introduction

Climate change mitigation pathways, created with integrated assessment models (IAMs), have come to play a critical role in the assessment work of Working Group III of the Intergovernmental Panel on Climate Change (IPCC) (Guivarch et al., 2022b; Riahi et al., 2022). The AR6 Scenarios Database, hosted by the International Institute for Applied Systems Analysis (IIASA), contains climate change mitigation pathways compiled for and considered in the Working Group III contribution to the IPCC Sixth Assessment Report (Byers et al., 2022; Kikstra et al., 2022).

In these pathways, carbon dioxide removal (CDR) from the atmosphere is primarily represented by bioenergy with carbon capture and storage (BECCS) and by carbon sequestration in land sinks – primarily via afforestation and reforestation (Riahi et al., 2022). Among the global scenarios in the AR6 Scenarios Database that passed the vetting process ($n = 1202$) (see Guivarch et al., 2022b, for details about the AR6 scenario vetting process), 419 pathways miss the variable for carbon sequestration on land ("Carbon Sequestration | Land Use"), which complicates secondary analyses that investigate CDR implications across scenarios and models. A

range of different secondary scenario ensemble evaluations based on data from the AR6 Scenarios Database have been published in recent years, for example, assessing the arising gap in CDR deployment (Lamb et al., 2024), determining the level and composition of residual emissions (Lamb, 2024), analyzing the removal per land unit (Zhao et al., 2024), evaluating the attainability of mitigation scenarios (Warszawski et al., 2021), classifying emission pathways reflecting the climate objectives of the Paris Agreement (Schleussner et al., 2022), or exploring scenario characteristics driving CDR deployment (Prütz et al., 2023). All of these analyses rely on proxy data or interim solutions to address the limited data availability of land carbon sequestration in the AR6 Scenarios Database.

Two such interim solutions to account for this data gap are documented in the literature, namely, (1) the use of net-negative $CO_2$ emissions in agriculture, forestry, and other land use (AFOLU) as a lower-bound proxy variable for CDR via land sinks (Prütz et al., 2023; Schleussner et al., 2022; Warszawski et al., 2021) and (2) a criteria-based scenario-filtering and exclusion approach to ensure a consistent selection of scenarios with similar reporting of CDR via land sinks (Prütz et al., 2023). Both approaches have limitations with respect to adequately and comprehensively depicting CDR via land sinks (Ganti et al., 2024). A more recent approach is based on a reanalysis of land $CO_2$ fluxes using the reduced-complexity compact Earth system model OS-CAR v3.2 (Gidden et al., 2023). While the AR6 reanalysis dataset by Gidden et al. (2023) manages to resolve several of the data issues linked to CDR via land sinks – specifically, aligning the removal baseline and improving the consistency across scenarios – it still combines gross and net $CO_2$ fluxes on land in the land sink CDR variable, resulting in both positive and negative CDR values, which conflicts with the concept and clean definition of anthropogenic CDR from the atmosphere (Matthews et al., 2021). In the AR6 Scenarios Database, CDR is conventionally reported using positive numbers. Moreover, while being very comprehensive, the re-analyzed dataset by Gidden et al. (2023) is limited to a subset ($n = 914$) of all global and vetted scenarios ($n = 1202$) of the AR6 Scenarios Database, although it also provides reanalyzed scenario data for five sub-global macro-regions (R5 level). Figure 1 compares the available land sink CDR data of the AR6 Scenarios Database to the reanalyzed variable by Gidden et al. (2023) and the net-negative AFOLU $CO_2$ proxy, showing the differences between the available land sink CDR data of the AR6 Scenarios Database, the net-negative AFOLU $CO_2$ proxy, and the land sink CDR data from the reanalysis. The figure also shows the negative values for land sink CDR from the reanalysis.

Here, we test and compare the performance of several different regression models to impute missing data on land carbon sequestration (Land CDR) based on available data on net $CO_2$ emissions in AFOLU for both global scenarios and the R10 regions in the AR6 Scenarios Database. We use the best-performing regression model to impute missing data for 404 global scenarios and 2358 sub-global scenario variants across the R10 regions and provide two imputation datasets, which are made publicly available. Lastly, we discuss our approach's use cases and limitations and detail how our approach compares to the two abovementioned interim solutions and the recent reanalysis of the AR6 Land CDR data. In the following, we refer to CDR via land sinks or carbon sequestration on land as "Land CDR". Table 1 gives an overview and description of key variables in this analysis.

## 2  Methods

### 2.1  Overview

In our analysis, we used different regression models to predict missing data on AR6 Land CDR (target variable: "Carbon Sequestration | Land Use") for 404 incomplete global scenarios and for 2358 incomplete sub-global scenario variants across R10 regions based on available scenario data on AFOLU $CO_2$ emissions (predictor variable: "Emissions | CO2 | AFOLU"). AFOLU $CO_2$ emissions were chosen as a predictor variable due to good data availability in the AR6 Scenarios Database and because this variable is conceptually most closely related to AR6 Land CDR among the variables in the AR6 Scenarios Database – the variable for AFOLU $CO_2$ emissions represents the net $CO_2$ fluxes corresponding to the gross variable for Land CDR, as defined in Table 1. The AR6 R10 region classification comprises 10 macro-regions plus 1 additional region for "rest of the world" (Fig. 4b), resulting in a total of 11 macro-regions, all of which were considered in our analysis. While the AR6 R10 classification allows for a comparison of regions across models and scenarios, not all regions are available for all models and scenarios, e.g., only a small subset of models has the category "rest of the world" (R10ROWO). For our analysis, we used the exact R10 regional classification as assigned in the AR6 Scenarios Database without excluding or adjusting regions for individual scenarios or models.

As an initial step, we selected all vetted scenarios from the AR6 Scenarios Database for which both the predictor and the target variable are available at the global level ($n = 783$) and across the R10 regions ($n = 6162$). Among the vetted global scenarios ($n = 1202$) in the AR6 Scenarios Database, 15 scenarios from the REMIND model (version 1.6) do not report AFOLU $CO_2$ emissions, which is why we could not include these scenarios in our imputation. Among the vetted regional scenario variants ($n = 8531$) across the R10 regions in the AR6 Scenarios Database, 11 regional variants of the EN_INDCi2100 scenario from the GEM-E3 V2021 model do not report AFOLU $CO_2$ emissions, which is why we could not include these scenario variants in our imputation. Figure 2 provides a simplified conceptual overview of the scenario selection, exclusion, and imputation workflow.

**Table 1.** Overview of the analysis variables.

| Variable | Description |
| --- | --- |
| "Carbon Sequestration \| Land Use" | This variable from the AR6 Scenarios Database is defined as the "total carbon dioxide sequestered through land sinks (e.g., afforestation, soil carbon enhancement, biochar)". This is the target variable that we impute for incomplete scenarios. In this analysis, we refer to this variable as AR6 Land CDR. |
| "AR6 Reanalysis \| OSCARv3.2 \| Carbon Removal \| Land \| Direct" | This variable from the reanalysis by Gidden et al. (2023) is intended to depict CDR through land sinks, similar to the AR6 Land CDR. However, the baseline $CO_2$ flux substantially differs compared with the AR6 Land CDR, as the data were aligned to national greenhouse gas inventories. This variable contains both positive and negative values, which suggests that it is showing net instead of gross removal. In this analysis, we refer to this variable as Gidden et al. (2023) Land CDR (direct). |
| "Emissions \| CO2 \| AFOLU" | This variable from the AR6 Scenarios Database is defined as the net "$CO_2$ emissions from agriculture, forestry and other land use (IPCC category 3)". This is the predictor variable that we use to predict the target variable. In this analysis, we refer to this variable as net AFOLU $CO_2$ emissions. |
| \| "Emissions \| CO2 \| AFOLU" $< 0$ \| | This variable shows the net $CO_2$ removal from agriculture, forestry, and other land use, based on the negative values in the variable net AFOLU $CO_2$ emissions. This variable has been used in several studies as lower-bound proxy for AR6 Land CDR. We refer to this variable as net-negative AFOLU $CO_2$. |
| "Imputed \| Carbon Sequestration \| Land Use" | This is one of two variables in the imputation datasets provided in this analysis. This variable contains the predicted values from our data imputation without further adjustment. |
| "Imputed & Proxy \| Carbon Sequestration \| Land Use" | This is one of two variables in the imputation datasets provided in this analysis. This variable contains the predicted values from our data imputation. For scenarios in which the predicted Land CDR is lower than the net-negative AFOLU $CO_2$, we replaced all predicted removals with the values from the net-negative AFOLU $CO_2$ and indicated this adjustment. |

We split both the global and the regional scenario datasets into training and testing sets (9 : 1) for our regression analysis in order to obtain a large dataset for training the models while also still having a sufficiently large testing dataset to evaluate the prediction performance and to validate the models. The training set was used to fit the predictor variable to the target variable to train the regression models, and the testing set was then used to evaluate the prediction performance of the trained regression models. The regression models were separately trained on the global scenario data and the regional scenario variants, as the scale of AR6 Land CDR deployment differs substantially between the global and the regional level. Regional scenario variants for model training were treated as one large training set, rather than splitting the data by R10 region before training. For both the global scenarios and regional scenario variants, we did not distinguish between AR6 scenario categories during the model-training process to keep the number of training data points as large as possible and, thus, optimize the models' performance. The AR6 scenario category (C1–C8) classification is based on the scenarios' global-warming levels from a low warming value of 1.5 °C with no or limited temporary temperature overshoot (C1) to a high warming value of more than 4 °C within this

century (C8) (Guivarch et al., 2022a). An overview of the AR6 scenario categories is provided in Table 2.

### 2.1.1 Regression models

We considered and compared four commonly used regression models in our analysis: gradient boosting, decision tree, random forest, and a $k$-nearest neighbors regression model. In the initial stage, a more extensive set of commonly used regression models, including linear regression and multilayer perceptron regression, were tested, among which the four abovementioned models were selected for further hyperparameter tuning due to their superior performance compared with other regression models in the initial set, based on the performance evaluation metrics described below.

For all models, we use the machine learning scikit-learn library for Python (Pedregosa et al., 2011). In the following, the four considered regression models are briefly described, while more detail is provided in the referenced seminal works and the scikit-learn documentation of the respective models, including the mathematical representations of the underlying algorithms. A decision tree model is a supervised learning method to predict a target variable based on decision rules derived from a predictor variable. The model produces piecewise approximations of the target variable through a series of

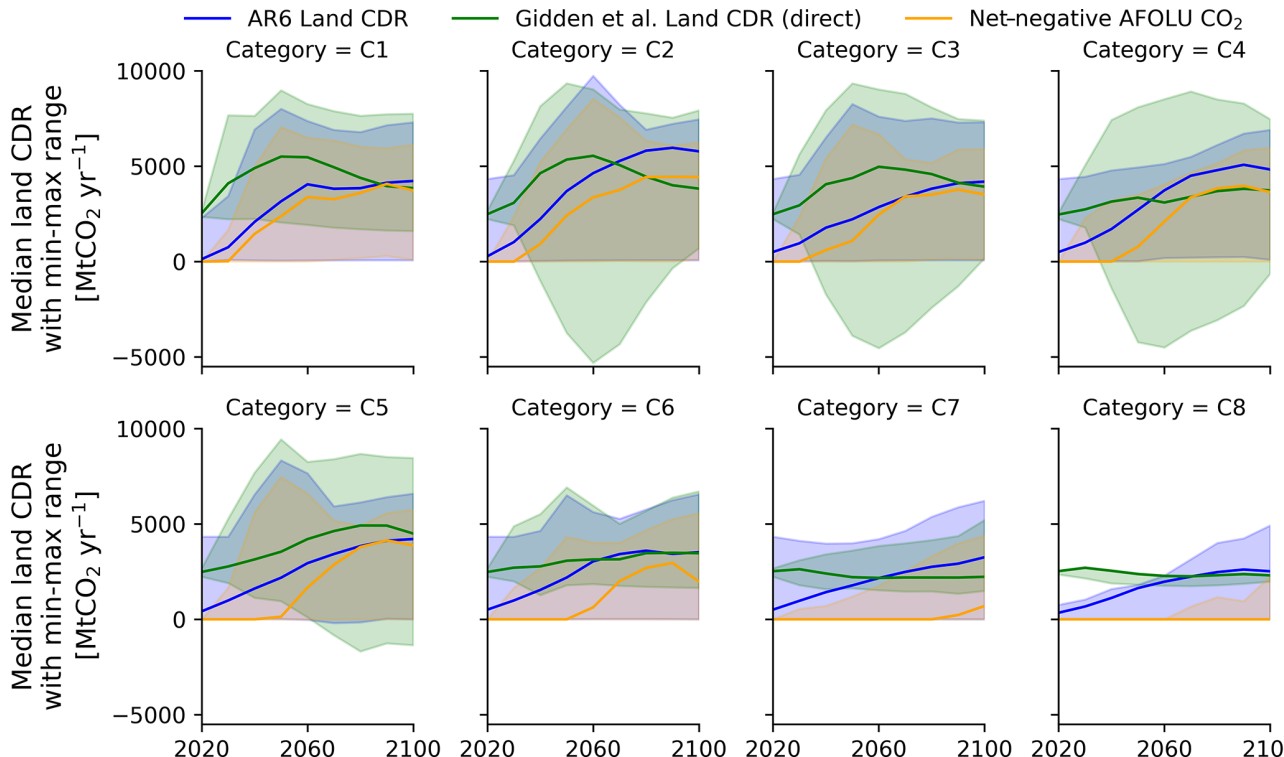

**Figure 1.** Comparison of available AR6 Land CDR data ("Carbon Sequestration | Land Use") with the Land CDR reanalysis by Gidden et al. (2023) ("AR6 Reanalysis | OSCARv3.2 | Carbon Removal | Land | Direct") and the AR6 net-negative AFOLU $CO_2$ emissions (based on negative values in "Emissions | CO2 | AFOLU") as a lower-bound proxy for AR6 Land CDR across AR6 scenario categories. Only scenarios available for all three variables are considered in the figure ($n = 725$ scenarios). The Land CDR scenarios in the reanalysis by Gidden et al. (2023) are aligned with national greenhouse gas inventories, shown by the difference in the baseline in 2020 compared with the other two variables. The solid lines show the median across scenarios, while the shaded area shows the min–max range. Note that we follow the convention of the AR6 Scenarios Database and report CDR using positive numbers, whereas the Land CDR variable in the reanalysis by Gidden et al. (2023) shows both positive and negative CDR numbers. An overview of the AR6 scenario categories (C1–C8) is provided in Table 2.

**Table 2.** Overview of scenario categories, as in Guivarch et al. (2022a).

| Category | Description |
| --- | --- |
| C1 | Scenarios limiting warming to 1.5 °C in 2100 (> 50 % probability) with no or limited overshoot (≤ 67 % exceedance probability of 1.5 °C) |
| C2 | Scenarios returning to warming of 1.5 °C in 2100 (> 50 % probability) after a high overshoot (> 67 % exceedance probability of 1.5 °C) |
| C3 | Scenarios limiting warming to 2 °C throughout this century (> 67 % probability) |
| C4 | Scenarios limiting warming to 2 °C throughout this century (> 50 % probability) |
| C5 | Scenarios limiting warming to 2.5 °C throughout this century (> 50 % probability) |
| C6 | Scenarios limiting warming to 3 °C throughout this century (> 50 % probability) |
| C7 | Scenarios limiting warming to 4 °C throughout this century (> 50 % probability) |
| C8 | Scenarios exceeding warming of 4 °C within this century (≥ 50 % probability) |

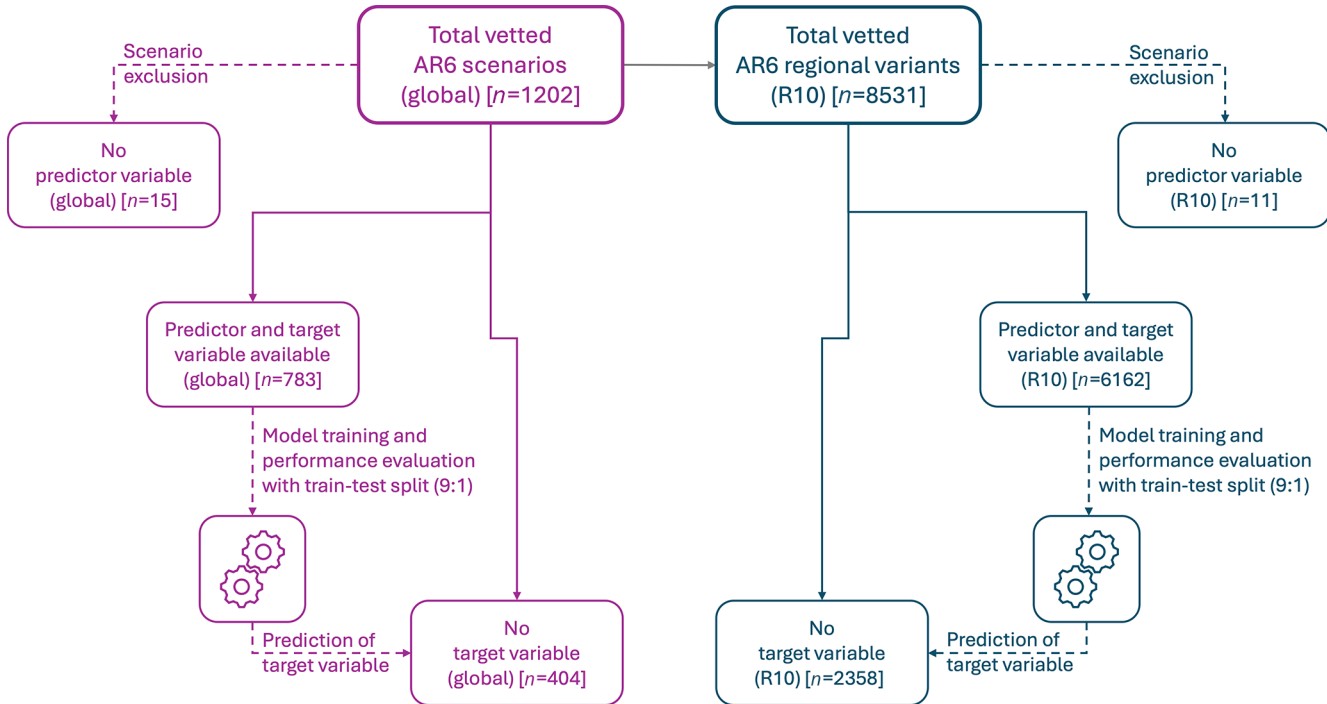

**Figure 2.** Conceptual overview of the scenario selection, exclusion, and imputation workflow for the global scenarios and for regional scenario variants at the AR6 R10 level. The numbers in parentheses indicate the respective number of scenarios. Dashed lines indicate a process, while solid lines depict the origin of a scenario subset. Details about the model selection, training, and performance evaluation are provided in Sect. 2.

binary data splits based on values of the predictor variable. For continuous predictor variables, such as net AFOLU $CO_2$ emissions in our case, the decision tree model iteratively selects thresholds for the predictor variable to split the data into a group above and below the respective threshold and then averages the target variable values per group to come up with a prediction. At each node of the tree, multiple potential splits are evaluated, and the imposed threshold that minimizes the prediction error is selected to increase the accuracy of the prediction of the target variable. This process continues recursively, splitting the data at each node until the tree reaches the leaf node, where no further splits are possible or no further reduction in the prediction error is achieved. At the leaf node, the tree makes a final prediction of the target variable – in our case, the expected AR6 Land CDR (Breiman et al., 1984). A gradient boosting regression model is an ensemble method that sequentially combines multiple simple models, called weak learners (typically decision trees, as described above), which correct the previous models' predictions to reduce the error and improve the final model (Friedman, 2001). A random forest model is also an ensemble method that combines multiple decision trees, as described above; however, unlike gradient boosting, the trees in a random forest model run in parallel instead of sequentially. Each decision tree works independently, and their individual predictions are averaged to produce the final prediction (Breiman, 2001). The

$k$-nearest neighbors model is not based on decision trees. Instead, it uses the proximity (similarity) of a scenario to a number ($k$) of neighbors (similar scenarios) to make predictions. For a given scenario, the model identifies the $k$-nearest data points of the predictor variable in the feature space and then averages the target variable values of these neighbors to come up with a prediction (Goldberger et al., 2004).

### 2.1.2 Performance optimization and evaluation

From the scikit-learn machine learning library, we used (1) grid search for our regression model hyperparameter optimization and (2) bootstrapping to estimate the variability in prediction performance for different subsamples of our training and testing data (Pedregosa et al., 2011). Grid search is an algorithm commonly used for regression problems and allows one to efficiently run regression models in different setups using all possible hyperparameter combinations to eventually select the best-performing model setup. The selection of hyperparameter options for the model optimization was driven by the observed model performance and computation time. We used bootstrapping to explore how the prediction performance of our optimized regression models varied based on different resamples ($n = 1000$) of the training and testing data, allowing us to better evaluate the robustness of the perceived performance of the tested models.

The model performance was determined based on four widely applied evaluation metrics, namely, $R^2$, mean absolute error, median absolute error, and maximum absolute error. These four evaluation metrics are briefly described in the following. $R^2$ was used to explore how well the tested regression model captured the relationship between the predicted variable and the actual variable in the validation dataset, indicating the goodness of their fit. $R^2$ can range from 0 to 1, with higher values indicating better fits. The other three evaluation metrics instead indicate absolute error, meaning the absolute difference between the predicted and the actual variable in the validation dataset throughout 2020–2100 – lower error values indicate more accurate predictions of the regression models. As the absolute error differs across variable pairs of the global scenarios ($n = 79$) and the regional scenario variants ($n = 617$) in the testing dataset, we reported the mean, median, and maximum error across the considered scenarios. Mean and median error are useful to estimate the prediction models' overall performance, whereas the maximum error is used to indicate the extreme in lower-end performance, based on the most inaccurately predicted scenario. Ultimately, the best-performing model ($k$-nearest neighbors) was used to impute the missing AR6 Land CDR data for incomplete global scenarios ($n = 404$) and incomplete sub-global scenario variants ($n = 2358$) across the R10 regions in the AR6 Scenarios Database. The performance of the four considered regression models and the selection of the $k$-nearest neighbors model is discussed further in Sect. 3.

### 2.1.3 Data post-processing

For all imputed scenarios, the predicted target variable was compared to their predictor variable to identify cases where imputed CDR on land is smaller than the respective net-negative AFOLU $CO_2$ emissions, as this conceptual error was partly also perceived in the AR6 Scenarios Database. The two imputation datasets for the global scenarios and R10 regional variants contain two data sheets. The first data sheet contains unadjusted imputation outputs. In contrast, the second sheet accounts for the conceptual error described above by replacing conceptually inconsistent predictions with their respective net-negative AFOLU $CO_2$ emissions for all years in affected scenarios to provide a lower-bound proxy for AR6 Land CDR – the implications of this are explained in Sect. 5. The code to implement the analysis and the global and regional imputation datasets are publicly available at https://doi.org/10.5281/zenodo.13373539 (Prütz et al., 2024).

## 3 Results

We show the performance of the four tested regression models along the four above-described evaluation metrics based on the testing set used for regression model validation for the global scenarios (Fig. 3a) and their regional scenario variants (Fig. 4a). Overall, the $k$-nearest neighbors regression model performs best, as it resembles the actual variable most accurately, while keeping mean, median, and absolute difference between the predicted variable and the actual variable comparatively low throughout 2020–2100 for both the global scenarios and the regional scenario variants. It also shows relatively low variance in performance across the bootstrapping results (Figs. 3a and 4a). While the gradient boosting regression performs comparatively well for the prediction of the global scenarios and slightly better concerning the maximum absolute error (Fig. 3a), the $k$-nearest neighbors regression outperforms the gradient boosting regression regarding mean and median absolute error for the prediction of the incomplete regional scenario variants (Fig. 4a). While the overall performance of these two regression models is similar, the $k$-nearest neighbors model was chosen to produce the two imputation datasets of this study, as the gradient boosting model partly predicted slightly negative values in the target variable, which is conceptually inconsistent with a clean definition of Land CDR, which should have a uniform removal sign. The other two regression models perform less well than the $k$-nearest neighbors and gradient boosting regressions. Overall, all models show a slight performance drop for $R^2$ around 2020–2060, with more stable or increased performance thereafter – we have found no convincing explanation for this slight temporal variation in performance.

In absolute terms, the mean, median, and maximum errors are larger for the evaluated global scenarios than for their regional scenario variants – this is expected due to the substantially higher levels of Land CDR deployment at a global level compared with the R10 regions. At a global level, the mean error of the $k$-nearest neighbors model is consistently below $200\,\mathrm{MtCO_2}\,\mathrm{yr^{-1}}$, while the median error is consistently below $40\,\mathrm{MtCO_2}\,\mathrm{yr^{-1}}$; at the R10 region level, we see a mean error consistently below $15\,\mathrm{MtCO_2}\,\mathrm{yr^{-1}}$ and a median error close to $0\,\mathrm{MtCO_2}\,\mathrm{yr^{-1}}$. At both the global and regional level, the mean and median absolute differences between the predicted variable and the actual variable are judged to be reasonably low, based on the $k$-nearest neighbors regression, and the absolute difference between the actual and predicted variable is substantially smaller than between the actual variable and the net-negative AFOLU $CO_2$ emissions as a lower-bound proxy for comparison (see also Figs. 3b and 4b). As a point of reference, the median AR6 Land CDR deployment across available scenarios of the scenario categories C1–C8 in the AR6 Scenarios Database is $1253\,\mathrm{MtCO_2}\,\mathrm{yr^{-1}}$ for 2020–2060 and $3570\,\mathrm{MtCO_2}\,\mathrm{yr^{-1}}$ for 2060–2100; across the R10 regions, median deployment is 39 and $179\,\mathrm{MtCO_2}\,\mathrm{yr^{-1}}$, respectively. This means that the median error accounts for around 1 % of the median AR6 Land CDR deployment throughout the time series for the original global scenarios, while this value is even lower for the original regional scenario variants in the regression validation dataset. However, while the regression model seems to perform well overall based on the regression evaluation

dataset, the observed maximum error suggests substantially worse performance in extreme cases, when looking at the scenario with the highest absolute error (Figs. 3a and 4a).

Figures 3b and 4b show Land CDR across the global scenarios and their regional scenario variants in the regression validation dataset, considering the actual variable for AR6 Land CDR, the predicted Land CDR using the $k$-nearest neighbors regression, and the net-negative AFOLU $CO_2$ emissions as a lower-bound proxy for comparison. Considering the scenarios in the global and regional regression validation datasets, the predicted variable appears to be a better proxy variable for missing AR6 Land CDR than the net-negative AFOLU $CO_2$ emissions proxy, as the predicted variable better resembles the shape of the actual variable and shows less absolute error throughout 2020–2100. While the predicted variable resembles the actual variable well across all eight AR6 scenario categories, Fig. 3b suggests some variance in performance across these categories – the drop in resemblance of the actual variable is most visible for C8 scenarios. This is at least partly due to the small number of underlying scenarios of this category in the regression validation dataset at the global level ($n = 2$). The prediction performance across the different R10 regions is comparatively consistent, as shown in Fig. 4b. In some cases, the actual variable and the predicted Land CDR are smaller than the net-negative AFOLU $CO_2$ emissions proxy (e.g., as visible in some instances for the R10 region R10EUROPE in Fig. 4b). This highlights a conceptual error in the underlying data, as further discussed in the subsequent section.

## 4  Code and data availability

The analysis code and the global and regional imputed datasets are publicly available at https://doi.org/10.5281/zenodo.13373539 (Prütz et al., 2024).

## 5  Discussion and conclusions

In this study, we tested and compared four regression models to impute missing AR6 scenario data on Land CDR based on available data on net AFOLU $CO_2$ emissions. The tested $k$-nearest neighbors regression model performed best and was used to impute the missing AR6 Land CDR data for incomplete global scenarios ($n = 404$) and incomplete sub-global scenario variants ($n = 2358$) across the R10 regions. The global and regional imputation datasets are publicly available at https://doi.org/10.5281/zenodo.13373539 (Prütz et al., 2024).

While we effectively resemble and impute AR6 Land CDR data for incomplete scenarios, our imputed datasets do not resolve underlying inconsistencies in the reporting of AR6 Land CDR in the AR6 Scenarios Database. The original data in the AR6 Scenarios Database for the "Carbon Sequestra-

tion | Land Use" variable are based on different reporting methodologies across IAMs, and land $CO_2$ fluxes are not always consistently and explicitly split into net-negative $CO_2$ emissions and gross removals (Ganti et al., 2024; Prütz et al., 2023). Different baselines for today's land removal are also perceived across scenarios, as shown in Fig. 1. For several scenarios in the AR6 Scenarios Database, net-negative AFOLU $CO_2$ emissions are larger than the reported AR6 Land CDR; this indicates conceptual errors, as Land CDR is a gross variable that can only be larger than or equal to net-negative AFOLU $CO_2$ emissions (Byers et al., 2022; Prütz et al., 2023). The issues of inconsistent removal baselines and net-negative $CO_2$ emissions being larger than gross removal are partly also perceived in our imputed datasets, as we use data from the AR6 Scenarios Database to train our model.

To address the latter problem, we provide an unadjusted imputation dataset as well as an adjusted imputation dataset for which we replaced conceptually inconsistent predictions (those with net-negative $CO_2$ emissions larger than gross removal) with their respective net-negative AFOLU $CO_2$ emissions for all years in the affected scenarios to provide a lower-bound proxy for AR6 Land CDR in the global and regional imputation dataset. We adjusted 106 global and 1594 regional scenario variants and indicated (in the adjusted imputation dataset) the scenarios for which the adjustment was made.

We emphasize that our global and regional imputed datasets are imperfect and that the persisting issue of net-negative $CO_2$ emissions and gross removals on land not always being separated or consistently reported across models must be considered when using our data imputation. Here, the applied approach to infill missing data is purely based on statistical relationships and is not intended to replace further improvements in comprehensively reporting Land CDR in the next generation of mitigation scenarios produced with process-based models. Nevertheless, Figs. 3b and 4b show that our imputed Land CDR variable is a markedly better proxy than the use of net-negative $CO_2$ emissions, which have partly been used in previous studies (Prütz et al., 2023; Schleussner et al., 2022; Warszawski et al., 2021) – both in terms of resembling the removal curve and reducing absolute error. Our imputation is also a better alternative to omitting a large part of the scenario space that does not report AR6 Land CDR.

Concerning use cases, we believe that our global and regional imputed datasets on AR6 Land CDR are most useful for analyses that aim to use the largest possible set of both original and imputed global scenarios ($n = 783 + 404$) or regional R10 scenario variants ($n = 6162 + 2358$) and a uniform carbon removal sign. Such scenario ensemble assessments are relevant to better understand a range of different aspects concerning Land CDR in climate change mitigation pathways. Several specific use cases have been highlighted above, including an assessment of the arising gap in CDR deployment (Lamb et al., 2024), an analysis of residual emis-

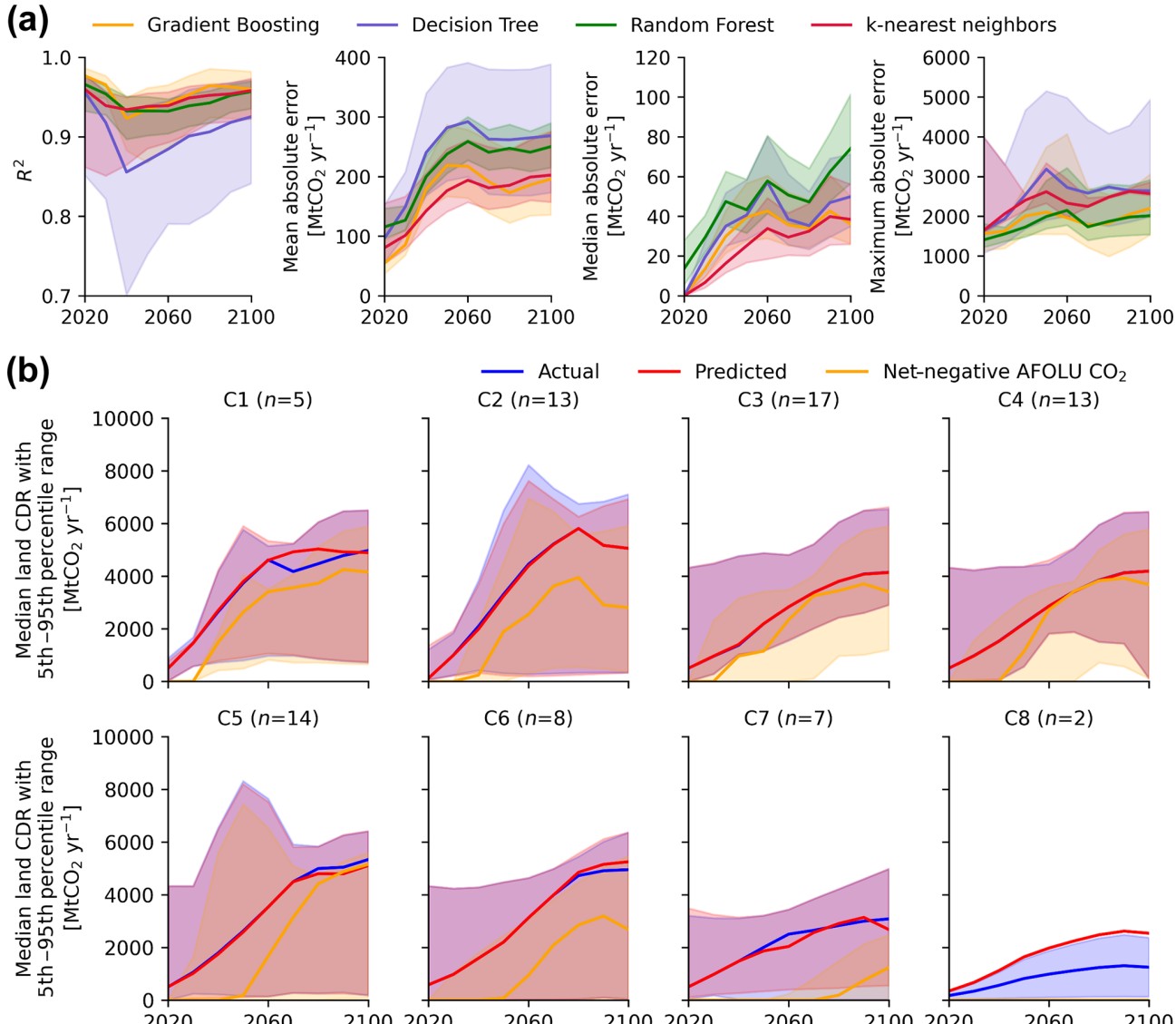

**Figure 3.** Prediction performance for the global scenario data. Panel **(a)** shows the performance of tested regression models to predict missing AR6 land removal data based on the used regression validation dataset ($n = 79$ scenarios). Performance across the four evaluation metrics is shown as the median (solid line) and 5th–95th percentile range (shaded area) of the bootstrapping results ($n = 1000$) for each of the four tested regression models. The performance results refer to the comparison between the predicted variable and the actual variable in the regression validation dataset. Panel **(b)** shows the actual ("Carbon Sequestration | Land Use") versus predicted Land CDR and the AR6 net-negative AFOLU $CO_2$ emissions (based on negative values in "Emissions | CO2 | AFOLU") as a lower-bound proxy for AR6 Land CDR across AR6 scenario categories in the regression validation dataset ($n = 79$ scenarios). The predicted data in the figure are based on the $k$-nearest neighbors regression. The solid lines show the median across scenarios, while the shaded area shows the 5th–95th percentile range. Note that we follow the convention of the AR6 Scenarios Database and report CDR using positive numbers. An overview of the AR6 scenario categories (C1–C8) is provided in Table 2.

sions including the land sector (Lamb, 2024), estimations of land per removal (Zhao et al., 2024), or evaluations of the attainability of mitigation scenarios, which rely on Land CDR (Warszawski et al., 2021).

Thus far, such analyses have relied on insufficient proxy data or interim solutions to address the limited data availability of land carbon sequestration in the AR6 Scenarios

Database and could benefit from the more comprehensive dataset on Land CDR across scenarios provided in this work. Based on the evaluation of mean, median, and maximum absolute error of the regression model used here, it is advisable to use our dataset for analyses that rely on a large ensemble of scenarios, for example, all scenarios of a certain scenario category or even several categories. This is because

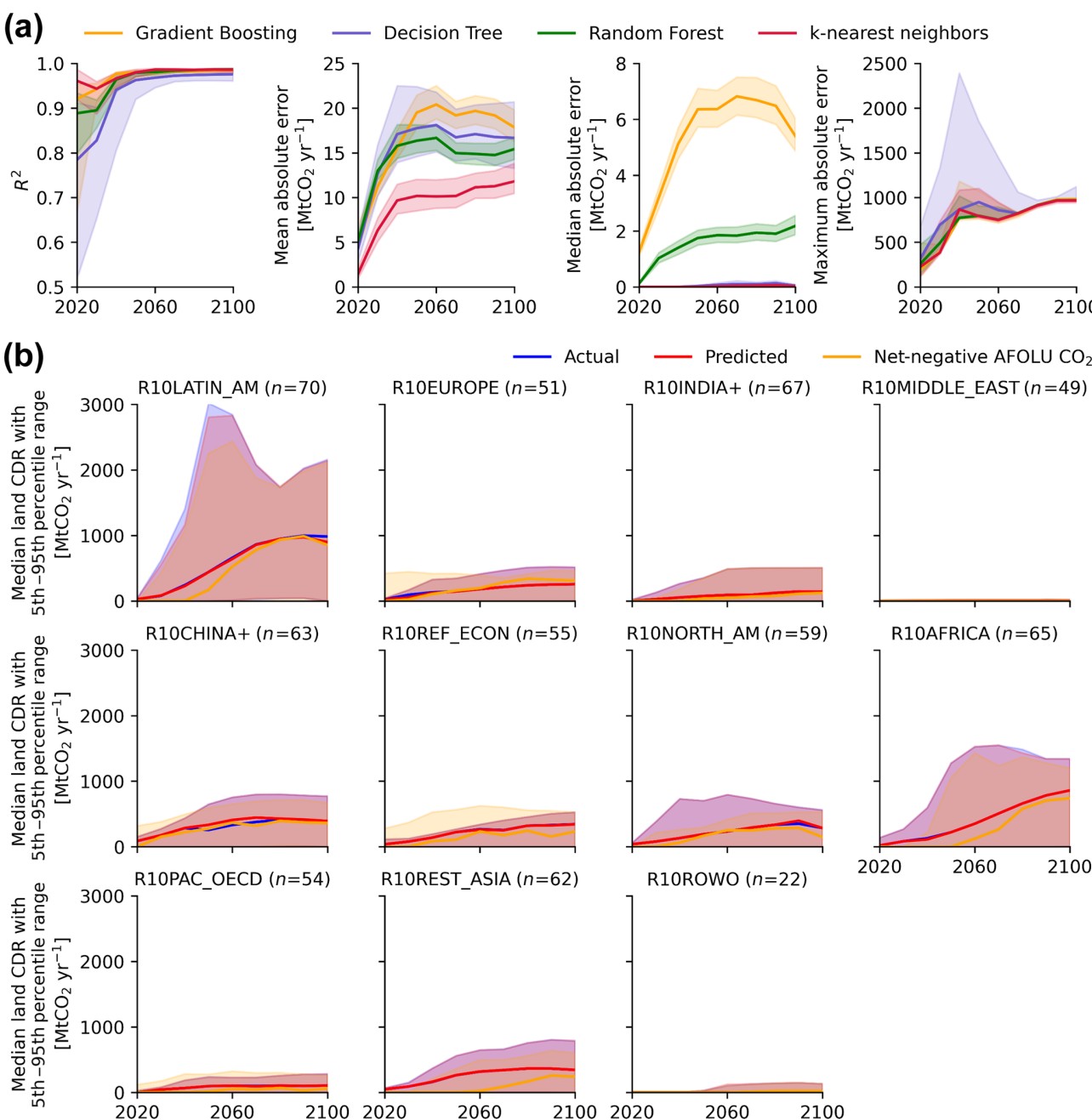

**Figure 4.** Prediction performance for the R10 regions' scenario data. Panel **(a)** shows the performance of tested regression models to predict missing AR6 land removal data based on the used regression validation dataset ($n = 617$ for regional scenario variants). Performance across the four evaluation metrics is shown as the median (solid line) and 5th–95th percentile range (shaded area) of the bootstrapping results ($n = 1000$) for each of the four tested regression models. The performance results refer to the comparison between the predicted variable and the actual variable in the regression validation dataset. Panel **(b)** shows the actual ("Carbon Sequestration | Land Use") versus predicted Land CDR and the AR6 net-negative AFOLU $CO_2$ emissions (based on negative values in "Emissions | CO2 | AFOLU") as a lower-bound proxy for AR6 Land CDR across AR6 R10 regions in the regression validation dataset ($n = 617$ for regional scenario variants). The predicted data in the figure are based on the $k$-nearest neighbors regression. The solid lines show the median across scenarios, while the shaded area shows the 5th–95th percentile range. Note that we follow the convention of the AR6 Scenarios Database and report CDR using positive numbers.

the prediction results from the regression model are more reliable for scenario ensembles than for individual scenarios, which may show larger error, as shown by the maximum error in Figs. 3a and 4a – arguably, it is generally advisable to aim to use scenario ensembles instead of individual scenarios to better capture uncertainties and diverse underlying assumptions, which may lead to more robust and credible analysis outcomes (Guivarch et al., 2022b). The reanalysis discussed above by Gidden et al. (2023) is perceived to be more suitable in terms of the consistency and accuracy of today's removals and for direct comparisons of scenario data and national greenhouse gas inventories (NGHGI). While our imputation dataset contains Land CDR data for the base year 2020, present-day and historical emissions and removals are better captured and more comprehensively discussed by the Global Carbon Project (Friedlingstein et al., 2023) – the merit of our imputation dataset lies in the future time steps of scenarios. As the imputation approach used in this work is purely based on statistical relationships between the predictor and target variable, it can also be applied to data availability problems in other domains to infill missing data, given that sufficient data are available to train and evaluate the model and that the models' performance is judged to be adequate. Ultimately, we hope that this study can be a valuable and complementary addition to the existing approaches addressing the Land CDR data gap in the AR6 Scenarios Database.

**Author contributions.** RP led the study and conceptualization, with supervision from SF and JR. RP implemented the analysis and wrote the original manuscript. All authors reviewed and edited the paper.

**Competing interests.** The contact author has declared that none of the authors has any competing interests.

**Disclaimer.** Publisher's note: Copernicus Publications remains neutral with regard to jurisdictional claims made in the text, published maps, institutional affiliations, or any other geographical representation in this paper. While Copernicus Publications makes every effort to include appropriate place names, the final responsibility lies with the authors.

**Acknowledgements.** The authors wish to thank the Integrated Assessment Modeling Consortium (IAMC) and the International Institute for Applied Systems Analysis (IIASA) for their valuable work collecting and hosting quantitative integrated assessment scenarios from and for the research community. We also thank the scikit-learn team for providing the free software machine learning library that made this analysis easily implementable. The authors acknowledge funding from the European Union's Horizon 2020 Research and Innovation program under grant nos. 101003687 (PRO-VIDE) and 951542 (GENIE). Ruben Prütz acknowledges funding from the European Union's Horizon 2020 Research and Innovation program under grant agreement no. 101081521 (UPTAKE). Joeri Rogelj acknowledges funding from the European Union's Horizon 2020 Research and Innovation program under grant agreement no. 101003536 (ESM2025). Sabine Fuss acknowledges funding from the German Ministry for Education and Research under grant agreement no. 01LS2101F (CDRSynTra).

**Financial support.** This research has been supported by the Horizon 2020 program (grant nos. 101003687, 951542, 101081521, and 101003536) and the German Ministry for Education and Research (grant no. 01LS2101F).

**Review statement.** This paper was edited by Martina Stockhause and reviewed by Thomas Bossy and three anonymous referees.

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
