# Peer review of "Imputation of missing land carbon sequestration data in the AR6 Scenario Database"

_Earth System Science Data, 2024_

## Referee Comment (RC4)

**Review of –"Imputation of missing IPCC AR6 data on land carbon sequestration"**

**Summary-** The AR6 database and its results are widely used to understand and analyze future climate mitigation and adaptation pathways. Since some IAMs used in the AR6 database model/report net land use emissions rather than gross, data on carbon sequestered on land is often missing. In this paper, the authors conduct an imputation/statistical interpolation to calculate data on carbon sequestered on land for the AR6 database where not reported. Rather than developing a method to convert the net land use to gross, the authors adopt a statistical approach which directly calculates the carbon sequestration from land.

 The land carbon component is indeed a key component of the database. Therefore, I find the work of the authors important. However, the paper as presented seems to be related to a specific method of interpolation applied an existing set of data rather than the creation of a generally usable dataset (which would put this paper outside the scope for a data journal like ESSD) (**See Major comments 2, 4**). Moreover, I had several questions and concerns regarding the general applicability of these methods outside of interpolating the current state of the AR6 database (**See Major comments 1,2,3**). Moreover, as a minor but important point- while the length of the paper does not automatically equate with quality (and the paper is clearly written), I found that the paper is too brief in its explanations regarding its variables, methods (As a simple example, it contains no methods section neither in the main text or the supplementary file which is usually critical for a journal like ESSD) (**See Major comment 5**). Please see all detailed comments below. **Given the comments, I recommend rejection and resubmission at a later date**. However, this is subject to the editor's discretion. All my comments are meant as constructive and in good faith to my colleagues in the field.

**Major comments-**

1. **Scale of analysis**- One of the fundamental questions I had was whether this imputation exercise is conducted at a global scale or at a regional scale? The actual dataset released contains only the global results (https://zenodo.org/records/10696654). Also, from the text, I interpreted all results as global? i.e., the independent variable is the carbon sequestered on land globally? If this is the case, this seems to be a shortcoming of the approach since it ignores regional heterogeneity. All IAMs included in the AR6 database produce regional emissions results. An imputation such as this seems to consolidate all the underlying regional dynamics to a regression which I find unconvincing. This is especially true in an age when studies are focused on deriving fine resolution (pixel level) results from regional ones. Can the authors comment on this? How valid would this method be if applied to the regional scale? Also see comment no 3 below. If regional results are calculated, kindly discuss them in the manuscript (See sub-comments of comment no 5).

2. **General applicability of this method**- Since both the dependent and independent variables are coming from the same version of the AR6 database, it seems to me that this

method really just begins and ends with the current state of the AR6 database. The IAMs underlying this database are constantly evolving and several of these IAMs are now focused on developing gross emissions pathways from the land sector. If a new version of the AR6 database were to be released, would this method still be valid? If the answer is no, then the data created and presented here is not a dataset with general usability. Rather , it is just a method of extension of the current AR6 database.

3. **Applicability beyond AR6-** Related to the above point, the Global Carbon Project (GCP) which releases its carbon budget analysis that is annually published in ESSD (https://essd.copernicus.org/articles/15/5301/2023/), includes land use emissions and carbon sequestration historically, globally. In fact, in more recent versions, there are also national emissions and sequestrations included. Would the current method produce reasonable results for a database such as the GCP's? Since the GCP is an equally well known dataset/exercise with a richer historical dataset, I would recommend that the current method be tested on that dataset to justify general usability of this method and data. Note that the national and global land use emissions and sequestration numbers are also available here- https://www.globalcarbonproject.org/carbonbudget/archive.htm

4. **Interpolation of existing data or new data**- Given the above points, I believe rather than a generalizable imputation, the authors have rather presented an interpolation method for existing data on the AR6 database. While this is not an issue, this would classify this paper as a methods paper rather than the development of original and novel data. In which case, this paper is not a good fit for the given journal which is generally meant for data descriptors. Finally, I would describe the methods of the authors as an interpolation rather than an imputation. That seems like a more precise description of the methods in the paper.

5. **Lack of description of methods (and results)**- While the paper provides a summary of the methods, the, they are never really described in any detail in the manuscript. As a simple example- ESSD papers generally contain a detailed methods sections which describe and justify the methods underlying the data which increase usability and reproducibility. This paper does not describe the method used in much detail which I find concerning. I have added specific comments related to the same below-

    **i)** Can the authors show a scatterplot of the x and y variables underlying the R squared values shown in Figure 2? Can these be separated for the training and the testing dataset?

    **ii)** What are the actual equations used for each of the methods/models? What were the final coefficients chosen for the gradient boosting method?

    **iii)** I checked the AR6 database, and it seems regional results are available. Therefore, related to comment 1 above, is the regression trained on global or regional results? If it's trained on regional results, can the regional heterogeneity be discussed in the manuscript? Does the regression fit all regions in the same way?

    **iv)** How and why is the current independent variable (IV) chosen? Were other IVs tested? Does performance change when different IVs are chosen?

**v)** The selection of the four final models for hyper parameter testing are described in a few short sentences. Can the details regarding the selection of the four models be added to the paper or supplementary information?

**vi)** I had trouble understanding Figure 3. What are the categories described in the figure? Can the authors describe what each of the categories represent? Why do C8 category emissions differ so much from observed compared to other categories? Was training and testing differentiated by category?

---

## Author Response (AR1)

**Point-by-point response:** https://doi.org/10.5194/essd-2024-68

| Reviewer comment | Author response |
|---|---|
| **RC1** | **AC1** |
| Review of Prütz et al: | |
| The manuscript presents a new dataset that imputes carbon removal on land for 404 incomplete scenarios in the AR6 Scenario Database. The authors identify a gap in the existing literature for overcoming missing data in the AR6 Scenario Database and propose a solution to fill it. Although their method is imperfect, the dataset is clearly presented and easy to use. Therefore, it deserves to be published. | Thanks a lot for taking the time to provide these thoughtful and constructive comments — they are much appreciated! Below, we respond to the general and specific comments point by point. |
| General comments on the dataset: | General comments: |
| | Regarding a merged dataset of imputed and original scenarios: |
| The dataset uses the same model and scenario names as in the AR6 database, making it very easy to use as a supplement for someone willing to fill in the missing data. However, as the authors themselves acknowledge, the Gidden et al. re-analysis is "considered more useful in terms of consistency and accuracy". Therefore, I see the interest of their dataset mainly for someone who needs land carbon sequestration for as many scenarios as possible. In that case, why not provide a dataset with all scenarios, and not just those where 'Carbon sequestration\|Land use' is missing? This would make it even more convenient for everyone to work with all scenarios in the same file, and it would provide an adjusted version that also corrects inconsistencies found in the original database, with net removals being greater than gross removals. | Thank you for this very useful suggestion! We have now adjusted our imputation dataset to include both the imputed and the original complete scenarios from the AR6 Scenario Database, which allows users to easily work with our dataset without having to first merge our data with the AR6 data. In the dataset, we clearly specified which scenarios were imputed and which were directly taken from the AR6 Scenario database. |
| Since the Gidden et al. reanalysis seems to be of better quality, would it be possible to add a paragraph discussing the feasibility/relevance (or not) of applying the same reanalysis to the imputed scenarios? | Regarding the reproducibility of the approach by Gidden et al.: |
| | To our understanding, the OSCAR-based approach by Gidden et al. requires more comprehensive land use change data per scenario (compare Gidden et al. 2023) and is therefore restricted to 914 scenarios, which meet the data requirements. Our alternative approach does not have these requirements and can, therefore, be applied to a larger set of scenarios (n=783+404) for the global level. Beyond the number of imputed scenarios, our approach does not allow for both positive and negative CDR values to be closely aligned with a clean |

| | conceptual definition of gross CDR – this differs from the OSCAR-based approach, as shown in Figure 1. Please consult the updated discussion section for details |
|---|---|
| Specific comments: | Specific comments: |
| l35. "While the AR6 re-analysis dataset by Gidden et al. manages to resolve several of the data issues linked to carbon removal on land, it still combines gross and net CO2 emissions on land in their land-based CDR variable, resulting in both positive and negative CDR values, which conflicts with the concept and clean definition of gross CDR." | Regarding the positive and negative CDR values in Gidden et al.: |
| | As this phenomenon is only shown by some models and scenarios in the reanalysis by Gidden et al., we suspect that this is at least partly driven by the properties of the original database. However, from the information at hand, we cannot say this with certainty. Gidden et al. would be better equipped to explain the underlying dynamics of the OSCAR model that might explain the conceptually unintuitive coexistence of both positive and negative "gross" CDR values. |
| Is it related to the fact that it is already inconsistent in the original Database or is it due to the re-analysis itself? | |
| Figure 1. | Regarding Figure 1: |
| Would it be possible to add one sentence explaining why do re-analysis look so different than the other 2 curves? In particular, the mismatch between the points in 2020. | The mismatch in data points in 2020 is due to different emission baselines, which have or have not been aligned across scenarios. This is partly already described in the original manuscript (see lines 124, 35-42). We again highlighted this in the caption of Figure 1 in the revised manuscript. |
| l71-74. After reading carefully, I think I understood how the metrics were calculated. However, it was quite fuzzy at first. Would it be possible to add an illustrative figure showing what the four metrics correspond to? | Regarding the evaluation metrics: |
| | Thanks for sharing this reflection. We have now substantially updated the methods section and improved the description of the evaluation metrics. |
| l107-108. "Figure 3 suggests some variance in performance across these categories – for C8 scenarios, the drop in resemblance of the actual variable is most visible" | Regarding figure 3: |
| Isn't it because the independent variable is mostly null or near zero for C8 (and C7) scenarios that the prediction is bad? | Generally, our regression approach seems to also reasonably impute values based on near zero net-negative AFOLU $CO_2$ emissions. However, we agree that the very low levels of net-negative AFOLU in C7 and especially C8 may at least partly explain the drop in prediction performance, while emphasising that our prediction is still substantially more accurate in scale and shape resemblance than the formerly used net-negative AFOLU proxy. We will add a note on this to reflect on it in the revised manuscript. |
| l120-121. "our imputed dataset does not account for perceived land sequestration related data issues in the AR6 Scenario Database beyond data availability" I find the phrasing unclear. | Regarding the unclear paragraph: |
| | L120-121 means to highlight that our approach cannot resolve underlying inconsistencies in the original dataset and mainly addresses the issue of lacking data availability. We rephrased the sentence in the revised manuscript to make it clearer. |
| **RC2** | **AC2** |

This paper uses statistical methods to fill in missing data on land carbon sequestration in 404 scenarios of the AR6 Scenario Database. The purpose seems useful, and the concept is generally clear. However, the study remains rather superficial in several instances, for example regarding the testing of the estimated regression models, describing the methods, and placing the study within the existing scientific literature.

Major comments:
1. The gradient boosting methods is suggested to be the best method. However, I miss a rigorous testing of this result, especially given that this method does not perform best in all of the analyzed metrics. I strongly advocate for using bootstrapping to estimate the variability in the analyzed metrics for the different regression methods, particularly to test whether the gradient boosting method is not just by chance the best method. Also, the selection of the best method should be clearly supported by numbers rather than by visual inspection.

2. The methodological description is currently too vague to be reproducible. A few questions that came to my mind: How were the data selected/pooled? Were all scenarios and all years pooled or just scenarios or years? Are all scenarios/values global averages or do they provide spatially explicit estimates? In case of the latter, how was this considered? How was the dataset split into training and testing sets? How was the superior performance (line 67) of the four selected regression models tested? Further details should be added, such that other scientists would be able to repeat the analysis.

   I also believe that a better justification for correlating gross removals with net fluxes is needed, as the two might not necessarily be correlated, for instance in cases were the temporal variability of net fluxes is dominated by emissions (and not removals).

We want to thank anonymous referee #2 for taking the time to review our manuscript. Below, we respond to the major and specific comments point by point and outline how we plan to revise our manuscript based on the feedback provided.

Major comments:
Response to major comment 1: We appreciate the suggestion to use bootstrapping to estimate the variability in performance for different subsamples of our training data. We have now included bootstrapping (n=1000) to evaluate the robustness of our models for different subsamples of the training and testing data. Further, we are using Grid Search and cross-validation to identify the optimal hyper-parameter combination of the tested regression models.

Response to major comment 2: In terms of reproducibility, we want to emphasise that both the input data as well as our analysis code is made available, allowing other researchers to fully trace and re-run our analysis. However, we appreciate the feedback on the level of detail in the written method section and have now improved the methodological description so the method can be more easily understood without having to consult the code.

Regarding the sub-questions under comment 2: As described in lines 57-60 in our original manuscript, we selected all vetted scenarios from the AR6 Scenario Database for which both the independent and the dependent variable are available (n=783) at the global level. Among the vetted scenarios (n=1202) in the AR6 Scenario Database, 15 scenarios from REMIND 1.6 do not report AFOLU $CO_2$ emissions, which is why we could not include these scenarios in our imputation. For each of these 783 scenarios, we select all years between 2020-2100 in 10-year steps. The data is directly downloaded from the AR6 Scenario Database and limited to the globally aggregated data - spatially explicit (gridded) scenario data is not part of the AR6 Scenario Database. The scenario data of the 783 scenarios was randomly split into train and test sets (using 1000 bootstraps for resampling the splits) using the scikit-learn library in Python. Initially, we also

included linear regression and multilayer perceptron regression but excluded these models as they performed substantially worse compared to the other four models based on the same evaluation metrics presented in Figure 2 and 3. This was also described in lines 65-76 of our orignial manuscript.

1. I feel like the introduction could and should provide more background why this study is relevant. For instance, some use cases of CDR estimates from the AR6 Scenario Database could be cited. I believe that this would also increase the visibility of this study.

   Related to this, the paper mentions that use cases for the approach would be presented, but I did not really find them in the manuscript (except for one rather generic sentence in the discussion). Being more specific here would be useful.

Response to major comment 3: Regarding the net fluxes as predictor for our model - we agree that net emission fluxes are not ideal due to the mentioned variability in the variable driven by other factors than removals. However, since gross emissions are not available, we see the net emission fluxes as the best available option. We addressed this valid comment in the discussion section by better reflecting on the implications of our imputation, which is based on statistical patterns rather than physical relationships. We also highlighted a few use cases of CDR estimates, which are based on the AR6 Scenario Database, to highlight its relevance and the contribution of our imputation to such analyses. The use cases of our dataset, as well as its advantages and disadvantages compared to other approaches, are already highlighted in lines 130-141 of the original manuscript.

Specific comments:
- Title: I would suggest checking if the word "imputation" is the best term in the context of this study. I am no native speaker, and when checking the translation of imputation I found that it can have several meanings, which might cause misunderstandings. Maybe "completion", "amendment" or something similar might be clearer. Also I think that changing the title to " [Imputation/amendment] of missing data on land carbon sequestration in the AR6 scenario database" would be a more adequate title

Specific comments:
Title: We feel that imputation as a term works well in the context of our regression-based data imputation. However, we appreciate the suggestion to restructure the title for better readability and propose: "Imputation of missing land carbon sequestration data in the AR6 Scenario Database". We have changed the title accordingly.

- I suggest being more specific in the abstract on what "gross carbon removal on land" means. Does this only refer to anthropogenic influences or does it also include the natural land sink? I suggest to check throughout the manuscript to make this clear (e.g. in line 28 as well)

Comment on gross carbon removal: We appreciate the feedback and did check the manuscript for consistency concerning the use of key terms. Generally, we mean anthropogenic removal only and exclude natural fluxes. However, it is not always clear how consistently this is done across scenarios, as also described in our manuscript (see abstract – line 12). We have also added Table 1 which provides an overview of key variables in the analysis.

| | |
|---|---|
| • Line 12: I read this sentences several times, but it did not really become clear what this means ("net and gross removals are not separated" and "consistently reported"). I suggest being more specific here. | Line 12: The mentioned sentence highlights that there is currently no consistency across scenarios in AR6 in terms of how much $CO_2$ is stored in land sinks. While some models explicitly distinguish between gross removal (CDR via afforestation and soil carbon sequestration) and net removal (net negative emissions in the AFOLU sector), other models only report the latter. There is also a spread in baselines of gross removal across scenarios, which suggests inconsistent definitions. |
| • Line 13: I would suggest to better link this sentence to the previous ones by making it clear that carbon removal estimates are essentially the gross anthropogenic removals. | Line 13: Thanks! We have now rephrased the abstract. |
| • Line 14: Is "net CO2 emissions" the right term? It sounds surprising if one wants to deduce removals from emissions. Also, the net flux is not necessarily an emission but could also be a removal. What about "net CO2 fluxes"? | Line 14: "Net emissions" or "Net $CO_2$ emissions" are well-established and commonly used terms. Net emissions are simply the sum of gross positive emissions and gross negative emissions. Indeed, "net emissions" can be negative if removals from the atmosphere exceed emissions to the atmosphere. "Net $CO_2$ flux" would work as well. |
| • Abstract: I'd suggest to add a sentence at the end of the abstract about potential implications or use cases of the study | Comment on use cases in the abstract: Thanks for this feedback. We slightly rephrased the abstract and also gave specific examples of use cases in the introduction of the manuscript. |
| • Line 29: I do not really understand the difference between the "net negative CO2 emissions in AFOLU" and the variable "Carbon Sequestration/Land Use". As this differentiation is necessary to understand the study, I think this needs to be better explained. | Line 29: We appreciate this feedback. The understanding of this distinction is indeed crucial for understanding our study. Simply put, "net negative $CO_2$ emissions in AFOLU" refer to the net removals on land while "Carbon Sequestration\|Land Use" refers to the gross removals (CDR) on land. We have now added Table 1 which describes the differences between the variables used in our study. |
| • Line 33: I think that a bit more information about the filtering and exclusion of scenarios in Prütz et al. (2023) would be useful here. | Line 33: In Prütz et al. (2023), an additional vetting process is applied based on a set of criteria which leads to scenario exclusion if scenarios do not meet the defined criteria. We have now added text and briefly elaborated on this in the revised manuscript. |
| • Line 36: "issues linked to carbon removals on land": I suggest giving a few examples to make this clearer (this also relates to my major comment 3 above) | Line 36: This sentence refers to the previously described issues of: lacking data availability, inconsistent baselines for land sinks, and inconsistent definitions of removals on land or the distinction between net and gross. We made sure to better link this to previous sections in the manuscript. |
| • Lines 40-41: What exactly is meant by "net-negative AFOLU | Line 40-41: "Net-negative AFOLU $CO_2$ proxy" means that the negative values in the net variable for AFOLU |

| | |
|---|---|
| CO2 proxy"? Does this refer to what is mentioned in lines 30-31? I suggest to use consistent terminology throughout the document to avoid misunderstandings. | $CO_2$ emissions of the AR6 database are used as substitute variable, i.e., proxy for missing data on land carbon sequestration (CDR via afforestation or soil carbon sequestration). We improved the consistency of our terminology and introduced the new table 1, which gives an overview and description of key variables in our analysis. |
| • Lines 40-42: What are the implications of this discrepancy? | Line 40-42: The discrepancy highlights the caveats of the approaches to resemble carbon sequestration (CDR via afforestation and soil carbon sequestration): either large deviation from the actual values in the AR6 database or due to conflicts with the concept and clean definition of gross CDR. The implications are described in detail in lines 30-40 and later also in lines 103-105 and lines 127-136 of the original manuscript. |
| • Line 68: What is grid search? | Line 68: Grid Search is a common machine learning hyperparameter tuning technique to find the optimal combination of model hyperparameters – in our case the setting options of our four regression models. We have now updated the methodological description. |
| • Line 77-78: Why is this conceptually false? Shouldn't a removal be negative? Or is there another reason? | Line 77-78: Thank you for raising this clarification question. Removals are reported as positive values in the AR6 Scenario Database. Therefore, predicted negative values would mean emissions. We realize that this is not obvious to readers and added a brief explanation to our manuscript and also in the figure captions. |
| • Line 94-95: Where can this consistent behavior be seen? And how was this tested? | Line 94-95: As part of our revisions and the integration of bootstrapping, we highlighted how the different models differ in terms of their performance and the variability of the performance metrics used to evaluate the models' performances. This is shown in Figure 2 two for the global scenarios and in Figure 3 for the new R10 region scenarios. |
| • Lines 95-96: This could be a statistical error. Bootstrapping may help to test this. | Line 95-96: Thanks again, we have now included bootstrapping. |
| • Line 107-108: Any ideas why it does not work so good for the C8 scenario? And what exactly is this scenario "C8"? | Line 107-108: C8 is one of the eight scenario categories used by the IPCC in AR6 to group scenarios based on their warming outcome. This category includes a total of 29 scenarios that show warming levels above 4°C (>4°C peak warming with ≥50% chance). It is the category with the lowest number of scenarios which may partly explain why the performance is lower compared to the other scenarios. We have added a reference and brief the description for the scenario categories. |
| • Line 108: Do the regression-based estimate fill the full uncertainty | Line 108: I am trying to understand whether I fully understand this question. We use and train different regression models based on available data to predict |

| | |
|---|---|
| range just like the original estimates? Or I their uncertainty larger or smaller than the original estimates? | missing data. We test the prediction performance of the models by using train test splits to evaluate how accurately the models can predict the available data. The full performance is shown in Figure 2 for the global scenarios and in Figure 3 for the new R10 region scenarios based on four evaluation metrics. While imperfect, this approach allows for the reasonable resemblance of the shape and size of the target variable where missing. The shaded areas in the figures are not uncertainty ranges but show the spread on the underlying scenario sets (C1-8). In other words, the shades areas do not show the confidence interval but the 5-95 percentile range (from low to high scenarios). |
| • Line 124: What does this mean and to what does this sentence refer to? I cannot relate it to Figure 1 and also not to the results shown in the paper. | Line 124: In Figure 1, one can see how there is a clear spread in baseline land removal levels; in other words, the level of removal in the year 2020, taken as the current condition, is not the same across scenarios. For example, for the scenarios in category C2, the baseline removal level ranges from no removals as of today up to almost 5000 MtCO$_2$. The data by Gidden et al. does not entail such a spread as the baseline is harmonized. This spread refers to the inconsistencies in reporting that we highlight several times in the paper (e.g., line 12). |
| • Lines 129 and 130: I would suggest adding the exact numbers of scenarios (as done in other places as well) | Lines 129-130: Agreed, we have now included this number in the revised manuscript. We also better specified the number of underlying scenarios in Figure 2 and Figure 3. |
| • Line 133: I think it would be good to shortly mention the issues again to help readers remind them. | Line 133: Thanks for the suggestion. We rephrased the sentence in the revised manuscript. |
| • Line 136: Where is the reduction in absolute error shown? | Line 136: The reduction in absolute error is due to our statistical approach, which is more suitable to resemble the shape and size of gross removal values instead of simply using net negative emissions as a crude proxy for CDR. The reduction is clearly visible in Figure 2b for the global scenarios and in Figure 3b for the new R10 region scenarios (red is much more similar to blue than yellow is). |
| • Line 140: Add full reference for Gidden et al. | Line 140: Gidden et al. is already cited several times in previous manuscript sections and in the reference list - we don't use the publication year when referring to papers in line. We are happy to change this if advised by the editor team. |
| • Line 140-141: What does this imply? Should one skip the scenarios that do not have estimates of gross carbon sequestration in such cases? | Line 140-141: Our dataset allows users to use all scenarios. All scenarios can be included because we provide the imputation of the missing sequestration data for incomplete scenarios. This is the main contribution of our analysis to the community. We highlight that, in future analyses, where the largest possible set of scenarios and a uniform removal sign |

| | |
|---|---|
| | are not as relevant as a uniform baseline and direct comparability to the NGHGI, users may want to consider using the dataset by Gidden et al.. As described in our manuscript, we see different strengths and caveats of the two approaches depending on the aim of users (see lines 120-143 of the original manuscript). |
| • Figure 1: What are the categories? A description of them is missing. What exactly is meant by "conservative proxy"? | Figure 1: These are the scenario categories used by the IPCC in AR6 to group scenarios based on their warming outcome (Table II.7. Classification of global pathways into warming levels using MAGICC in Guivarch et al. (2022)). We have now highlighted this in the revised manuscipt.

"Conservative" in the context of "AR6 net negative AFOLU $CO_2$ emissions (based on negative values in 'Emissions\|CO2\|AFOLU') as a conservative proxy for land-based CDR across AR6 scenario categories." implies that it is a rather low approximation, underestimating the actual size of removals. We have now changed the wording to facilitate the understanding. |
| • Figure 2: I strongly suggest to not put the decision tree, random forest, and k-nearest neighbor more transparent than gradient boosting, as this might cause biased interpretations of the results (even if the black line is not the best one it might be considered the best one, just because it is less transparent). I also wonder whether the error estimates could additionally be shown as percentages of the mean flux estimates. Right now, it is difficult to say how large the errors are without knowing the mean flux estimates. Also, is this figure just for one scenario or were scenarios somehow pooled? I suggest adding that in the caption. | Figure 2: We now revised the figures and used uniform linewidth and transparency. The use of percentage values for the error assessment is not feasible because the dependent variable can contain zero values, especially for earlier years in the timeseries where CDR deployment has not started. Therefore, percentage values are not feasible as we cannot divide by zero to assess the percentage difference between the dependent and independent variable. It is also not advisable to use percentage error when evaluating the performance of prediction models because the error can be falsely high if the predicted values are close to zero – we therefore focus on four standard evaluation metrics. As described in the figure caption, this is based on 79 scenarios, i.e., 79 predictions in our validation dataset of the global scenarios, giving the overall R-squared, mean, median, and absolute error compared to the actual values of the scenarios in the validation dataset. |
| **RC3**

Information on the level of gross carbon dioxide removal (CDR) on land is incomplete in the AR6 Scenario Database. They are represented through the variable Carbon Sequestration/Land use, which is missing in 419 out of 1202 pathways. In this paper, the authors use the variable ,net negative CO2 emissions in agriculture, forestry, and other | **AC3**

We thank referee #3 for providing reflections and feedback on our manuscript. This is much appreciated! In the following, we respond to the major and smaller comments in detail and describe how we plan to revise our manuscript based on the reviewer's feedback. |

land use (AFOLU)' as a proxy variable for land-based CDR. Therefore, they test the performance of several regression methods. They find gradient boosting regression performs best using $R^2$, the mean absolute error, the median absolute error, and the maximum absolute error as metrics.

In general, this is a very valid and essential approach to improve estimates on land-based CDR. However, the description and explanations of the respective variables and the method are much too short and not detailed enough in my point of view. The authors should further extend the discussion on limitations and use cases of their approach to underline its relevance. To fully validate the soundness of the statistical approach, I recommend an additional review by an expert on statistics.

Major Comments:

The authors should describe in detail what the relevant variables they use are (Carbon Sequestration/ Land use, net negative CO2 emissions in agriculture, forestry, and other land use (AFOLU)), e.g. do they include C uptake from natural forests and would that even point to CDR as it only involves intential interventions in land use? Does the latter include wood products and emissions from land management? Differences between the two variables and shortcomings in the approach should be discussed in-depth. Maybe a figure would help demonstrate the gross and net fluxes of both variables.

Further, the description of the statistical approach should be elaborated more (mathematically and in words). To demonstrate the robustness of the methods, cross-validation or bootstrapping should be applied. Which metrics did the authors use to show that the applied methods are superior to the other methods? Could you briefly explain the statistical methods you use? I find it also questionable if the approach is valid for all categories (Fig.1) as e.g. for C7 and C8 the net negative AFOLU fluxes are 0 for most of the time. Doesn't it make sense to exclude specific categories?

Major comments:

Major comment 1: We appreciate the feedback on the description of the two variables that we are working with. We now included the brief variable descriptions from the AR6 database to clarify their scope further. Generally, natural C uptake should not be part of these two variables (gross and net) – only anthropogenic fluxes should be included. The AR6 database documents "Carbon Sequestration|Land Use" as "total carbon dioxide sequestered through land-based sinks (e.g., afforestation, soil carbon enhancement, biochar)". "Emissions|CO2|AFOLU" is defined as "$CO_2$ emissions from agriculture, forestry and other land use (IPCC category 3)". However, we also highlight apparent inconsistencies across scenarios – partly shown by the range in baseline fluxes (2020) for these variables. We highlight other approaches that address the scenario data issues concerning land emissions and compare them to our data imputation approach. We have now included Table 1 in the revised manuscript which describes the used variables and includes the documentation from the AR6 Scenario Database for the different variables.

Major comment 2: We welcome the suggestion to include bootstrapping in our evaluation of model performances, we have included this in our revised manuscript version. We already use hyperparameter tuning with Grid Search and cross validation to optimize the performance of our models (details are shown in our Zenodo repository). We evaluate the model performance based on four common evaluation metrics (namely, R-squared, mean absolute error, median absolute error, and maximum absolute error) and evaluate the performance for each time step in the time series (Figure 2). The added bootstrapping and the improved method section will further facilitate the

reading. We also want to highlight that all required data and code are made available – thus, our approach is fully tracible and replicable.

About the models used: For all models, we use the scikit-learn library for Python. The decision tree model splits the data of the independent variable into different branches based on the features of this variable (AFOLU emissions in different years). Each node in the decision tree represents a decision based on a feature, and each leaf node represents a prediction. The tree is then built by choosing the best feature splits to maximize the accuracy of the predictions. The gradient boosting regression model sequentially combines multiple simple models, called weak learners (typically decision trees), which correct each other's predictions to minimize the overall error and improve the final model. The random forest model also combines multiple decision trees, but unlike gradient boosting, these trees run in parallel instead of sequentially. Each decision tree works independently, and their individual predictions are averaged to produce the final prediction. The k-nearest Neighbors model instead uses the proximity (similarity) of a scenario to a number (k) of neighbors (other scenarios) to make predictions. For a given scenario, the model identifies the k nearest data points in the feature space and then averages the target variable values of these neighbors to come up with a prediction. We have now briefly described the models in the methods section of the revised manuscript.

Indeed, the prediction performance is lower for the scenario categories C7 and C8. However, please note that there are several scenarios in categories C7 and C8 for which there are net negative $CO_2$ emissions towards the end of the century (see the yellow shaded area in Figure 1). More importantly, for the training of the regression models, we did not only use the net-negative values of 'Emissions|CO2|AFOLU' but all values (also positive ones). Also, for the model training, we did not distinguish between scenario categories - this was only done for the data visualization.

| Smaller comments | Smaller comments: |
|---|---|
| • l. 33: scenario filtering and exclusion: please explain more in detail

• l. 33: Which limitations do they have? Name at least one for each approach. | Line 33: In Prütz et al. (2023), an additional criteria-based scenario vetting process is applied, which leads to scenario exclusion if scenarios do not meet the defined criteria. While this allows for a more consistent scenario selection, it reduces the number of scenarios in the set. This is a limitation, as we want to be able to look at all available scenarios to consider the largest set of potential pathways. Using the net-negative emissions as a proxy does not require scenario exclusion; however, it underestimates the real level of CDR on land as residual emissions on land are not zero. We adjusted the wording in the revised manuscript. |

| | |
|---|---|
| • l. 35: OSCAR v3.2 is not an earth system model but a reduced-complexity model with an explicit treatment of the land-use sector | Line 35: Thanks for pointing this out. The authors refer to it as a compact earth system model, but we are happy to emphasize its reduced-complexity and explicit treatment of the land-use sector. Changes were made accordingly. |
| • Fig. 1: explain categories → helps to understand discrepancies (e.g. Why are net negative AFOLOU $CO_2$ fluxes 0 for C7, C8?) | Figure 1: We used the scenario categories by the IPCC in AR6 to group scenarios based on their warming outcome (Table II.7. Classification of global pathways into warming levels using MAGICC in Guivarch et al. (2022)). We now linked back to the AR6 report which details the different scenario categories and also highlighted the source in the figure captions. Several scenarios in C7 and C8 show net-negative $CO_2$ emissions in AFOLU in the second half of the century. However, a larger share stays net positive throughout the century as residual emissions remain larger than removals in this sector for many scenarios. This is why the median in Figure 1 is so low (however, the yellow shade shows quite some net negative emissions towards the end of the century). For the regression analysis, this does not matter as the models are trained based on the original variable "Emissions\|CO2\|AFOLU", not just its negative values. |
| • l. 59: What is the fractional choice of the split in training and test sets based on? To prove the robustness, it makes sense to apply cross-validation here. | Line 59: We are using a large share (90%) of the available data for training to optimize the model and use 10% for testing. There is no fixed rule for how large the training and testing split should be, and 90:10 is perceived to be a common ratio. We already use Grid Search and cross validation to optimize the hyperparameter selection for our models. As part of the revisions, we included bootstrapping to resample the train test split many times (n=1000) to test the robustness of the model performance for different splits. This helped us to evaluate the models better. |
| • l. 71: Why do you need three metrics to evaluate the absolute difference between the dependent and the independent variable? | Line 71: Technically, we don't need three metrics. However, these are commonly used metrics to evaluate the performance of regression models. We include all three because we feel like this gives additional information and helps us understand the model's performance. For example, the median and mean absolute error show that the models overall well manage to predict the size of removals with acceptable error (only a few megatonnes). However, in extreme cases, the error of the prediction can be more substantial with gigatonne-scale error. The R-squared instead helps to understand how well the models manage to resemble the shape of the time series rather than the size of the error. |
| • l. 80: related to a more detailed explanation of variables: why is this a conceptual error? | Line 80: It is a conceptual error because gross negative emissions cannot be smaller than net negative emissions, as net emissions are the sum of gross positive and gross negative emissions. Assuming that there would be no residual emissions, gross negative |

| | emissions would equal net negative emissions, but their net negative emissions cannot be larger than gross negative emissions. We have now adjusted the text in the manuscript to make this clearer. |
|---|---|
| • Fig. 3: What does the shaded area show? Add to caption. | Figure 3: It is the 5-95 percentile range (from low to high scenario in the underlying scenario set), as indicated in the y-axis label. We have now reiterated this in the caption. |
| • l. 138: Could you add more detailed use cases here? | Line 138: Thanks for the suggestion; we added more tangible examples to the discussion and also the introduction to highlight the different benefits and caveats of the approaches. We also mentioned several examples of scenario assessments that can benefit from our imputation dataset in the introduction. |
| • l. 140: Why is the Gidden et al. Dataset perceived to be more consistent with today's removal? | Line 140: The Gidden et al. dataset is perceived to be more consistent with today's removals as it is aligned with the national greenhouse gas inventories and shows harmonized removals in the year 2020, whereas in AR6, there is quite a range in removal levels for 2020 (see blue shaded areas in Figure 1). |
| **RC4**

Review of –"Imputation of missing IPCC AR6 data on land carbon sequestration"

Summary- The AR6 database and its results are widely used to understand and analyze future climate mitigation and adaptation pathways. Since some Integrated Assessment Models (IAMs) used in the AR6 database model/report net land use emissions rather than gross, data on carbon sequestered on land is often missing for specific scenarios. In this paper, the authors conduct an imputation/statistical interpolation to calculate data on carbon sequestered on land for the AR6 database where not reported. Rather than developing a method to convert the net land use to gross, the authors adopt a statistical approach which directly calculates the carbon sequestration from land.
 The land carbon component is indeed a key component of the database. Therefore, I find the work of the authors important. However, the paper as presented seems to be related to a specific method of interpolation applied an existing set of data rather than the creation of a generally usable dataset (which would put this paper outside the scope for a data journal like ESSD) (See Major comments 2, 4). | **AC4**

We thank the anonymous referee #4 for providing detailed feedback on our manuscript. Below, we respond to the individual comments point by point. |

Moreover, I had several questions and concerns regarding the general applicability of these methods outside of interpolating only global results from the current state of the AR6 database (See Major comments 1,2,3). Moreover, as a minor but important point- while the length of the paper does not automatically equate with quality (and the paper is clearly written), I found that the paper is too brief in its explanations regarding its variables, methods (As a simple example, it contains no methods section neither in the main text or the supplementary file which is usually critical for a journal like ESSD) (See Major comment 5 for details). Please see all detailed comments below. Given the comments, I recommend rejection and resubmission at a later date. However, this is subject to the editor's discretion. All my comments are meant as constructive and in good faith to my colleagues in the field.

Major comments-
1. Scale of analysis- One of the fundamental questions I had was whether this imputation exercise is conducted at a global scale or at a regional scale? The actual dataset released contains only the global results (https://zenodo.org/records/106966 54). Also, from the text, I interpreted all results as global? i.e., the independent variable is the carbon sequestered on land globally? If this is the case, this seems to be a shortcoming of the approach since it ignores regional heterogeneity. All IAMs included in the AR6 database produce regional emissions results. An imputation such as this seems to consolidate all the underlying regional dynamics to a regression which I find unconvincing. This is especially true in an age when studies are focused on deriving fine resolution (pixel level) results from regional ones. Can the authors comment on this? How valid would this method be if applied to the regional scale? Also see comment no 3 below. If regional results are calculated, kindly discuss them in the manuscript (See sub-comments of comment no 5).

Major comments

Major comment 1: Indeed, our statistical approach was initially applied to the global outputs from the AR6 Scenario Database - therefore, our imputed variables are also global ("Imputed|Carbon Sequestration|Land Use" and "Imputed & Proxy|Carbon Sequestration|Land Use"). This is also clearly indicated in the column "Region" in the provided spreadsheet in the Zenodo repository, following the convention in the original AR6 Scenario Database. It is also specified in the original manuscript (line 39). We do not see our focus on the global level as a shortcoming of our analysis as the global variable is very often used in scenario assessments and, therefore, of great value for other researchers (for example, Lamb et al. 2024 ERL, Lamb et al. 2024 Nature, Prütz et al. 2023 ERC, Schleussner et al. Commun Earth Environ 2022).
We think including regional scenario data from the AR6 Scenario Database is an interesting addition to our global analysis. Therefore, we now also included an imputation dataset for the 10 macro regions in the AR6 Database (R10), as the regional data availability was sufficient to be included here.

| | |
|---|---|
| 2. General applicability of this method- Since both the dependent and independent variables are coming from the same version of the AR6 database, it seems to me that this method really just begins and ends with the current state of the AR6 database. The IAMs underlying this database are constantly evolving and several of these IAMs are now focused on developing gross emissions pathways from the land sector. If a new version of the AR6 database were to be released, would this method still be valid? If the answer is no, then the data created and presented here is not a dataset with general usability. Rather , it is just a method of extension of the current AR6 database which would put this paper outside the scope of ESSD. | Major comment 2: Indeed, our approach is intended to provide a complete dataset of missing data in the current version of the AR6 Scenario Database. We hope that in the future, this will not be required anymore as the reporting of variables on net and gross $CO_2$ emissions and removals on land becomes more consistent across models. This means that our manuscript is not intended to introduce a method of general applicability. Instead, we aim to provide a more complete scenario data set for carbon sequestration on land for researchers working on scenario assessments. Of course, the imputed scenarios have a shelf life (so does the AR6 Scenario Database in general), especially due to forgone emission reductions, which are making the ambitious scenarios more and more counterfactual. We still think that our imputation dataset is of great value as long as the current AR6 Scenario Database is in use. We want to emphasize that our contribution to the field is the provided dataset, not the method we used to infer the data. Researchers require comprehensive scenario data on carbon sequestration on land to be able to assess the role of CDR across scenarios comprehensively. Therefore, we believe that our provided dataset and manuscript is well within the scope of ESSD as the editor's decision to consider our manuscript for review also suggests. |
| 3. Applicability beyond AR6- Related to the above point, the Global Carbon Project (GCP) which releases its carbon budget analysis that is annually published in ESSD (https://essd.copernicus.org/articles /15/5301/2023/), includes land use emissions and carbon sequestration historically, globally. In fact, in more recent versions, there are also national emissions and sequestrations included. Would the current method produce reasonable results for a database such as the GCP's? Since the GCP is an equally well known dataset/exercise with a richer historical dataset, I would recommend that the current method be tested on that dataset to justify general usability of this method and data. Note that the national and global land use emissions and sequestration numbers are also available here- https://www.globalcarbonproject.or g/carbonbudget/archive.htm | Major comment 3: As highlighted in the previous section, our contribution to the field is the provision of an imputed dataset of land carbon sequestration for incomplete scenarios in the AR6 Scenario Database - not the statistical methods used to infer the data. We do not aim to provide a generalized methodology but to provide a useful and complete dataset for the research community. Theoretically, the tested machine learning models could also be used to infer missing data in pretty much any other incomplete dataset (given that there is enough data for training and validating the model) because the used method is based on statistical relationships and not physical dynamics specific to $CO_2$ fluxes on land. Therefore, it would possibly be also applicable to infer data for historical emissions, e.g., linked to the Global Carbon Project. However, this is not the scope and purpose of our manuscript, as we infer missing data for future mitigation pathways. |
| 4. Interpolation of existing data or new data- Given the above points, I | Major comment 4: We feel that imputation as a term works well in the context of our regression-based data |

| | |
|---|---|
| believe rather than a generalizable imputation, the authors have rather presented an interpolation method for existing data on the AR6 database. While this is not an issue, this would classify this paper as a methods paper rather than the development of original and novel data. In which case, this paper is not a good fit for the given journal which is generally meant for data descriptors. Finally, I would describe the methods of the authors as an interpolation rather than an imputation. That seems like a more precise description of the methods in the paper. | imputation. However, we are happy to use the term interpolation instead of imputation if this is perceived to be more accurate by the journal editors. However, we do not see how this affects the question of whether our provided dataset falls within the scope of the journal, and we want to kindly highlight that our manuscript would not be under review if the handling editor had concluded that our manuscript was not within the scope of the journal as part of the initial access review. |
| 5. Lack of description of methods (and results)- While the paper provides a summary of the methods, they are never really described in any detail in the manuscript. As a simple example- ESSD papers generally contain a detailed methods sections which describe and justify the methods underlying the data which increase usability and reproducibility. This paper does not describe the method used in much detail which I find concerning. I have added specific comments related to the same below- | Major comment 5: We thank the reviewer for their suggestion to further elaborate on the methods. We want to highlight that all required data and code to replicate the study are publicly available in the Zenodo repository making our analysis fully traceable. However, we appreciate the feedback and now added more detail to our existing methods section to allow for a better understanding of our approach, even without consulting the code underlying our analysis. As part of the revisions, we also included bootstrapping in our approach to estimate the variability in performance for different subsamples of our training data. We also included a brief descriptions for each of the four models that we tested. We also plan to better highlight the differences between the scenario categories. |
| | Minor comments |
| • Can the authors show a scatterplot of the x and y variables underlying the R squared values shown in Figure 2? Can these be separated for the training and the testing dataset? | Minor comment 1: The requested scatterplot has been provided via the ESSD Discussion page showing the data underlying R-squared in Figure 2 (in this case for the gradient boosting model). The scatterplot contains 711 data points (79 scenarios x 9 time steps). The x-axis shows the predicted values for each scenario (n=79) and year based on the gradient boosting model. The y-axis shows the actual values (the true values in the AR6 Scenario Database) for the 79 scenarios in our regression validation dataset. Color in the plot indicates the different years in the time series per scenario. The unit for both axes is $MtCO_2$ removed via land carbon sequestration. In Figure 2a for the global scenarios and in Figure 3a for the regional R10 scenarios in our manuscript, we report R-squared for each time step in the time series. |
| • What are the actual equations used for each of the methods/models? | Minor comment 2: All tested models are based on the scikit-learn library for machine learning in Python, |

| | What were the final coefficients chosen for the gradient boosting method? | where the models/algorithms are described: https://scikit-learn.org. Gradient boosting: https://scikit-learn.org/stable/modules/generated/sklearn.ensemble.GradientBoostingRegressor.html Decision tree: https://scikit-learn.org/stable/modules/generated/sklearn.tree.DecisionTreeRegressor.html Random forest: https://scikit-learn.org/stable/modules/generated/sklearn.ensemble.RandomForestRegressor.html K nearest neighbours: https://scikit-learn.org/stable/modules/generated/sklearn.neighbors.KNeighborsRegressor.html We have now improved the description and referencing of the models. |
|---|---|---|
| | • I checked the AR6 database, and it seems regional results are available. Therefore, related to comment 1 above, is the regression trained on global or regional results? If it's trained on regional results, can the regional heterogeneity be discussed in the manuscript? Does the regression fit all regions in the same way? | Minor comment 3: We have now complemented our global analysis with regional data based on the AR6 R10 regions and updated the description of how the global and regional scenarios data was used to train the regression models. |
| | • How and why is the current independent variable (IV) chosen? Were other IVs tested? Does performance change when different IVs are chosen? | Minor comment 4: At an initial stage, we also tested other variables, e.g., net $CO_2$ emissions (aggregate of AFOLU and Energy & Industry emissions). Eventually, we decided to work with the net $CO_2$ emissions from AFOLU as this variable yielded the best results and was conceptually most closely related to carbon sequestration on land among the available variables in the database. We updated the description in the revised manuscript. |
| | • The selection of the four final models for hyper parameter testing are described in a few short sentences. Can the details regarding the selection of the four models be added to the paper or supplementary information? | Minor comment 5: Thank you for pointing this out. We have now elaborated more on the methods as part of the revisions. Specifically, we briefly described the underlying concept of each of the four models. |
| | • I had trouble understanding Figure 3. What are the categories described in the figure? Can the authors describe what each of the categories represent? Why do C8 category emissions differ so much from observed compared to other categories? Was training and testing differentiated by category? | Minor comment 6: Thank you for this feedback. These are the scenario categories used by the IPCC in AR6 to group scenarios based on their warming outcome (Table II.7. Classification of global pathways into warming levels using MAGICC in Guivarch et al. (2022). We elaborated on this in the revised manuscript. |

| EC1 | AC5 |
|---|---|
| I have given you the opportunity to substantially revise your manuscript, taking up the reviewers' suggestions, esp.:

- provide a detailed description and critical discussion of the applied methodology, e.g. by comparing your dataset to alternative data like data of the GCB (why is this method chosen? how would the application of another method alter your dataset?)

- provide further evidence of the quality of the dataset, which is derived from scenarios with an inherent considerable uncertainty

- what is the added value of your dataset?

- who is the target user community (which will not use alternative regional datasets or the global data directly)?

Please keep in mind that ESSD is a scientific journal. Methodology, originality of the dataset and an added value to the community are all important criteria for the acceptance of your submission. | Thank you for the opportunity to submit a revised version of our manuscript. We have now substantially updated the analysis and improved the description in the text. We made the following major revisions:
• In addition to imputing missing data for global AR6 scenarios, we have now also developed an imputation dataset for incomplete sub-global scenario variants based on the AR6 R10 region classification, to complement the global data with more granular data for 11 macro regions. As suggested by reviewer 1, we have now combined both the imputed and the available scenarios in our dataset to further facilitate the use of the data.
• We have now included bootstrapping in our analysis to use various resamples (n=1000) of the train-test-split to better evaluate the performance and robustness of the considered regression models. This highlights the uncertainty in the performance of the used regression models. We have also improved the description of the regression model outputs and provided more context on removals across scenarios to facilitate the interpretation of the prediction errors.
• We have substantially revised and detailed the methodological description of our analysis. This includes descriptions of each of the four considered regression models; a better and more consistent description of the key variables and terminology used in our analysis, including an overview table to facilitate the read; as well as a better description of AR6 specifics such as the reporting convention of removals and the scenario categories.
• We have further detailed the current problem of insufficient data availability and quality on carbon dioxide removal via land sinks across scenarios in the AR6 Scenario Database and commonly applied proxies and interim solutions. This is important to underline the added value of our imputation datasets.
• We have also added specific examples of previous studies relying on Land CDR scenario data, to highlight potential use cases of our imputation dataset. We have also expanded the discussion and better explained in which cases our imputation dataset may be useful for other researchers, and when the use of other datasets, such as the reanalysis by Gidden et al. or data from the Global Carbon Project may be better suited. This is required to better specify use cases and the target group of our imputation datasets. |

---

## Referee Report (RR1)

Referee Report:

We thank the authors for addressing the majority of our comments by providing an overview of the used variables in Table 1, including bootstrapping for a more robust analysis, adding use cases to the discussion, describing the categories and the different regression models used.

However, there are some minor points that are not yet fully clear to me.
I wonder why in the recent manuscript, the k-nearest neighbor method performs better compared to gradient boosting as in the previous version? Is that due to including bootstrapping in your analysis?

Which overall metrics do you use to decide which regression model performs best for Fig. 2A and Fig. 3A? Is the decision purely based on visual comparison or do you include some weighting of the four metrics? There is no clear explanation in the results why the k-nearest neighbor method performs best.

It would ease understanding if you clearly describe the difference between Fig. 2a and 3a at the beginning of the results section.

---

## Author Response (AR2)

**Point-by-point response:** https://doi.org/10.5194/essd-2024-68

| Reviewer comment | Author response |
|---|---|
| **RC1**
We thank the authors for addressing the majority of our comments by providing an overview of the used variables in Table 1, including bootstrapping for a more robust analysis, adding use cases to the discussion, describing the categories and the different regression models used.

However, there are some minor points that are not yet fully clear to me. | **AC1**
We thank the reviewer for taking the time to re-evaluate our revised manuscript and for the constructive comments. Below, we respond to the remaining comments point-by-point. |
| I wonder why in the recent manuscript, the k-nearest neighbor method performs better compared to gradient boosting as in the previous version? Is that due to including bootstrapping in your analysis? | Generally, the two models perform comparatively well for the global scenarios. Correct, the k-nearest neighbors model performs slightly better than gradient boosting when including bootstrapping for the global scenarios – the difference gets clearer when looking at the regional scenario variants (see the new Figure 4a and the first paragraph in the Results section). |
| Which overall metrics do you use to decide which regression model performs best for Fig. 2A and Fig. 3A? Is the decision purely based on visual comparison or do you include some weighting of the four metrics? There is no clear explanation in the results why the k-nearest neighbor method performs best. | We did not weigh or aggregate the four evaluation metrics. As mentioned above, k-nearest neighbors and gradient boosting perform comparatively well for the global scenarios. For the regional variants, k-nearest neighbors outperforms gradient boosting. Another argument against gradient boosting is that this model partly predicted slightly negative values in the target variable, which is conceptually inconsistent with a clean definition of Land CDR, which should have a uniform removal sign. In lines 177-186 in the revised manuscript, we described why we selected the k-nearest neighbors model instead of gradient boosting. |
| It would ease understanding if you clearly describe the difference between Fig. 2a and 3a at the beginning of the results section. | Thank you for this feedback. We have slightly rephrased the first sentence of the Results section to make the distinction |

| | between the old Figure 2a and 3a clearer (now Figure 3a and 4a). |
|---|---|
| **RC2**
The authors have substantially improved the manuscript based on the reviewer comments. The description of methods is much clearer, the discussion has improved, and the full paper is more comprehensible.

In a few instances, I still had difficulties to fully understand the content, as I felt that some explanations are still missing. I thus have a few remaining points that should be considered before publishing the manuscripts.

Remaining points:
• Please explain the meaning of all categories C1-C8. These are central to the analysis, as they are used in two out of the three figures of the manuscript. The categories C1-C8 first appear in Figure 1 but are not explained in the caption. Their meaning should also be explained in the caption of Figure 1. | **AC2**
We thank the reviewer for the constructive and detailed feedback on our revised manuscript. This was very helpful to improve the manuscript further! Below, we respond to the remaining comments point-by-point.

We have now introduced Table 2 which specifies the meaning of all eight AR6 scenario categories. We have also pointed to Table 2 in the captions of the figures to facilitate the read. |
| • It remains a bit unclear whether the R10 regions are 10 or 11 regions (including rest of the world), and whether the imputation dataset has data for 10 or 11 regions. Please specify this more clearly. | The R10 regional categorization comprises 10 regions plus an additional category for rest of the world (see new Figure 4b). Most IAMs have no more than 10 regions but there is a handful of models with 11. The models were trained based on all available data without excluding any of the R10 regions – the same applies for applying the model to impute missing data. We have adjusted the wording in lines 84-88 to make this explicit. |
| • Line 79-87: I cannot follow how the numbers in this section are derived. It also remains unclear how the final number of scenarios (404 and 2358) is connected to the numbers listed in this section. A clearer derivation of the numbers would thus be helpful. | We impute 404 global and 2358 sub-global scenario variants, which lack data on Land CDR in the AR6 Scenario Database. To do so, we focus on the vetted and complete global scenarios (n=783) and vetted and complete regional scenario variants (n=6162). We have now adjusted the wording to make this more |

| | comprehensible. We have also introduced the new Figure 2, which provides a better overview of how the numbers were derived. |
|---|---|
| • Sentence in lines 93-96: This sentence is unclear to me. What are the training bins? Why are scenarios not split and have they been split for other analyses? (and if so how and why have they been split?) I think that more explanation is needed here for readers who are not familiar with the methodology. | This means that we did train the model separately based on global scenario data and on regional scenario variants. However, we did not further distinguish and separately train scenarios by AR6 scenario category or R10 region. This was done to keep the number of training data points as large as possible. We have now rephrased this section to better clarify what we did. |
| • The discussion mentions that in some cases net negative AFOLU CO2 yields larger CDR than the predicted Land CDR. This already becomes evident in Figures 2 and 3 (e.g. for category C5 in Fig. 2 or region EUROPE in Fig. 3). I think it would be good to already point to this behaviour in the results section (as it is a result of the study). | Thank you for this suggestion. We have now added two sentences to the results section (lines 226-229) and refer to the discussion. |
| Minor points:
 • Line 14: Maybe add "not" before "consistently reported" (for clarity). | Thank you for this suggestion. We have adjusted the text accordingly. |
| • Line 16: It is not yet clear what R10 means. I'd suggest adding a short explanation here. | We have now slightly adjusted the wording in the abstract to address this feedback. |
| • Line 76: Not only the R10 regions but also global, right? | The 404 refers to global and the 2358 refers to the R10 regions. We changed the wording to "for 404 global scenarios and for 2358 sub-global scenario variants across R10 regions" to make the separation between global and R10 clearer. |
| • Line 80: Here, the term "net emissions" confuses me. To me "emissions" denote CO2 fluxes from the land to the atmosphere, whereas CDR are "removals". A more neutral term including | Thank you for this reflection. We have now adjusted the wording and referenced Table 1 again to make it clearer. |

both emissions and removals could be "fluxes". If the paper uses "emissions" in a broader sense (i.e., including emissions and removals), this should be clearly stated.

| | |
|---|---|
| • Line 81: Shouldn't the reference point to Figure 3b? Also, I suggest adding a map that shows the regions, as there are so many different definitions for regions out there that their naming remains unclear unless a clear depiction of their extent is shown. | Very well spotted, thanks for pointing out this typo! We have changed it accordingly. We agree that a visual overview of the different regions would be desirable. However, as we state in the Methods section, the underlying models, which produced the scenarios don't always use the identical classification of regions. To our understanding, the R10 regions were assigned by colleagues of IAMC and IIASA during the post-processing and curation of the AR6 Scenario Database, which brings the different models and scenarios together. We hope that our revised text still allows for a better understanding. |
| • Line 170 and following: This only refers to the k-NN approach, right? Please specify. | Correct, we refer to the k-nearest neighbors. We have now made this explicit. |
| • Line 172 and following: Check the superscript for yr-1. | Thanks for spotting this typo! We have now corrected the superscript. |
| • Figure 3: The caption for panel (b) is the same as for Figure 2b, although the results are shown here for different regions and not for scenario categories. Please check. Also, how are the different scenarios aggregated to derive the regional timeseries shown in Figure 3b? | Thank you for pointing this out. We have adjusted the caption accordingly. The regional scenario variants across the R10 regions are not aggregated in the old Figure 3b (now Figure 4b). Figure 4b shows the median and 5-95 percentile range across the regional scenario variants for the different R10 regions. In the subplot titles, the number of underlying scenario variants in the regression validation dataset is specified. The total number of regional scenario variants in this dataset is 617 but the number of variants varies across regions. |
| • Lines 230-232: I suggest adding references to the figures, scenarios, or regions, where this is the case. | In our imputation dataset, we have specified for which scenarios the issue of net negative $CO_2$ emissions being larger |

| | than gross removal occurs. The issue of inconsistent removal baselines is more like a cross-scenario issue. We have adjusted the wording to highlight that we made the former issue explicit in our imputation dataset. |
|---|---|
| • Line 233-236: How is the data replaced in the adjusted dataset? Are only single years replaced or is the full time series replaced? Also, what do the numbers mean in lines 235-236? | The full timeseries is replaces as shown in the imputation dataset. We made this explicit in line 259. The numbers indicate the number of scenarios for which we performed the replacement in the adjusted dataset. |
| **RC3**
Thank you for your revisions. I thank the authors for addressing all my comments and expanding on the methods in particular (This helped me greatly in derstanding the dataset and I'm sure it will be helpful to the community). I also appreciate the regional analysis. I also went through the other reviewer's comments and appreciate the value this dataset adds to the community.

I would recommend one small technical correction before publication. While the methods described here have been applied to AR6 data, authors should acknowledge whether or not the methods posted here can be extended to other datasets (such as the GCP for example). I don't think its an issue if the methods described here cannot be extended at the moment, but a comment on the same would be helpful for the community in my opinion. | **AC3**
We thank the reviewer for the positive feedback on our revised manuscript. Below, we respond to the remaining suggestion.

The regression model-based imputation approach could also be adapted and applied to other datasets to infill incomplete variables given that sufficient training data is available to tune and evaluate the regression model. Generally, the merit of our data description paper is not the method itself but the generated dataset, which is of use for researchers working with AR6 scenario data and facing the issue of incomplete reporting of Land CDR. While our approach allows to work with a more complete set of scenarios, it is still a workaround to deal with lacking data availability. For the next generation of mitigation scenarios, the issue should ideally be addressed by improving the reporting of Land CDR across integrated assessment models. See lines 264-266 and 290-293 in the revised manuscript, where we flagged this. |